# A BAFF ligand-based CAR-T cell targeting three receptors and multiple B cell cancers

Derek P. Wong [1,11], Nand K. Roy[2,11], Keman Zhang[2], Anusha Anukanth [3], Abhishek Asthana[2], Nicole J. Shirkey-Son[4], Samantha Dunmire[4], Bryan J. Jones [5], Walker S. Lahr[6,7,8,9], Beau R. Webber [6,7,8,9], Branden S. Moriarity[6,7,8,9], Paolo Caimi[2,10] & Reshmi Parameswaran [2,10 ✉]

B cell-activating factor (BAFF) binds the three receptors BAFF-R, BCMA, and TACI, predominantly expressed on mature B cells. Almost all B cell cancers are reported to express at least one of these receptors. Here we develop a BAFF ligand-based chimeric antigen receptor (CAR) and generate BAFF CAR-T cells using a non-viral gene delivery method. We show that BAFF CAR-T cells bind specifically to each of the three BAFF receptors and are effective at killing multiple B cell cancers, including mantle cell lymphoma (MCL), multiple myeloma (MM), and acute lymphoblastic leukemia (ALL), in vitro and in vivo using different xenograft models. Co-culture of BAFF CAR-T cells with these tumor cells results in induction of activation marker CD69, degranulation marker CD107a, and multiple proinflammatory cytokines. In summary, we report a ligand-based BAFF CAR-T capable of binding three different receptors, minimizing the potential for antigen escape in the treatment of B cell cancers.

[1] Department of Pathology, Case Western Reserve University, Cleveland, OH, USA. [2] Division of Hematology/Oncology, Department of Medicine, Case Western Reserve University, Cleveland, OH, USA. [3] Division of Pediatric Hematology/Oncology, Angie Fowler AYA Cancer Institute, UH Rainbow Babies & Children's Hospital, Cleveland, OH, USA. [4] Luminary Therapeutics, Minneapolis, MN, USA. [5] Bio-Techne, Minneapolis, MN, USA. [6] Department of Pediatrics, University of Minnesota, Minneapolis, MN, USA. [7] Masonic Cancer Center, University of Minnesota, Minneapolis, MN, USA. [8] Center for Genome Engineering, University of Minnesota, Minneapolis, MN, USA. [9] Stem Cell Institute, University of Minnesota, Minneapolis, MN, USA. [10] The Case Comprehensive Cancer Center, Case Western Reserve University School of Medicine, Cleveland, OH, USA. [11] These authors contributed equally: Derek P. Wong, Nand K. Roy. ✉ email: rxp278@case.edu

Chimeric antigen receptor T cell (CAR-T) immunotherapy has been a great success in the treatment of liquid cancers, providing rapid and durable responses[1–5]. However, disease relapse often occurs in these patients. The main reasons for relapse are either CAR-specific antigen loss on cancerous cells or poor performance of CAR-T cells due to exhaustion or decreased persistence[6–12]. Antigen escape has been reported not only for CD19-directed CARs, but also for BCMA[13], EGFRvIII[14], and IL13Rα2[15], highlighting the drawback of CARs targeting a single tumor-associated antigen. Thus, there is a great need to identify alternative targeting strategies and CAR designs[16]. Targeting multiple markers on malignant cells is a promising method to combat antigen escape[6]. Due to the inherent binding nature of naturally occurring ligands, CARs composed of a ligand or receptor ectodomain may have the ability to bind multiple proteins. A small number of ligand- or receptor-based CARs have been tested clinically and have shown encouraging results[17,18], presenting an intriguing design approach to combat antigen escape.

Most CAR-T therapies are focused on singular targeting of CD19 or other limited antigens[19]. Such concentrated effort on a few targets may distract future innovation by diluting resources and participation of patients in clinical trials, thus urgently warranting identification and development of new CAR targets. One attractive target is the set of B-cell activating factor (BAFF) receptors. BAFF ligand is a critical B cell survival factor that binds three receptors: BAFF-R, TACI, and BCMA[20]. These receptors are expressed by mature B cells and in a wide range of B cell neoplasms[21–25]. The ability of BAFF to bind to multiple receptors may protect against antigen escape due to the decreased likelihood that cancerous B cells can evade cytotoxic CAR-Ts by downregulating BAFF receptors, since they are important for cell survival. Furthermore, in contrast to approved therapies directed against the pan-B cell marker CD19, CAR-Ts designed to target BAFF receptors may be considered a more selective approach to eliminate malignant B cells, due to their more limited expression during B cell development. Currently, the production of CAR-Ts is primarily achieved using viral transduction of transgenes into primary human T cells[26].

Despite the relative success of viral transduction, the manufacturing of human CAR-T products is expensive, complex, and associated with notable safety considerations, supporting the need for alternative methods[27–29]. Since the first human application of transposon-mediated gene therapy almost 10 years ago[30], a dozen more clinical trials are underway or have been completed, supporting transposon-based systems as a safe and stable gene transfer alternative with comparable or superior efficiencies to viral transduction[31].

Here, we report the development and validation of a BAFF ligand-based CAR-T cell product, which can be successfully generated using the non-viral *TcBuster* (*TcB*) transposon system. We show that these BAFF CAR-T cells are both functional and specific in targeting the three BAFF receptors (BAFF-R, BCMA, and TACI) expressed by multiple B cell cancers. We demonstrate robust in vitro and in vivo cytotoxicity exerted by BAFF CAR-T cells against mantle cell lymphoma (MCL), multiple myeloma (MM), and acute lymphoblastic leukemia (ALL) xenograft mouse models.

## Results

**A BAFF ligand-based CAR-T developed using non-viral gene delivery**. We designed the BAFF ligand-based CAR construct using intracellular CD28, OX40/CD134 costimulatory domains, and the CD3ζ signaling domain, as well as the CD28 transmembrane domain (Fig. 1a). To effectively bind the BAFF

receptors, we used a truncated BAFF sequence that encompasses the majority of the extracellular domain of the natural human BAFF ligand. We also created a construct that lacks BAFF but is identical to the rest of the BAFF-CAR construct to serve as a negative control (no-BAFF control) (Fig. 1a). Due to the homology between soluble endogenous human BAFF and the extracellular domain of our BAFF-CAR, it is reasonable to predict that BAFF CAR-Ts will bind to the classical BAFF receptors BAFF-R, BCMA, and TACI, thus conferring multi-antigen specificity. Importantly, we also utilize a non-viral genetic engineering approach to stably integrate the BAFF-CAR into T cells using the *TcB* transposon system. Following co-electroporation of *TcB* transposase-encoding mRNA and transposon plasmid into T cells, the transposase enzyme excises the BAFF-CAR "cargo" from the transposon plasmid before integrating it into genomic DNA, from which BAFF-CAR protein is stably expressed (Fig. 1b). For traditional lentiviral transduction, a modified pLVX expression vector was used (Supplementary Fig. 1a). Both lentiviral and *TcB*-based delivery resulted in stable BAFF-CAR expression, as evidenced by the detection of BAFF ligand on the T cell surface via flow cytometry (Fig. 1c). Surface BAFF expression directly correlates with GFP expression in the lentiviral transduced T cells, as the pLVX vector encodes GFP downstream of the same promoter as the BAFF-CAR construct. The CD4/CD8 ratios of donor T cells were also characterized (Fig. 1c). T cells transduced with the no-BAFF control construct express GFP, but not surface BAFF (Supplementary Fig. 1b). BAFF CAR-T cells produced using the *TcB* transposon method were also assessed for stable expression of BAFF-CAR. After 7 days of expansion, BAFF CAR-T cells produced from different donors were frozen, then thawed and cultured for 21 days. CAR-T cells at 21 days post-thaw expressed a similar amount of cell surface BAFF as those 5 days post-thaw (Supplementary Fig. 1c).

Next, we conducted preliminary cytotoxicity experiments against the Jeko-1 MCL cell line to confirm BAFF-CAR function. T cells were co-cultured with fluorescently-labeled Jeko-1 cells at a 5:1 effector:target (E:T) ratio for 16 h, then stained with propidium iodide (PI) and gated on the cancer cells to determine cancer cell death. BAFF CAR-T cells exhibited significantly higher cytotoxicity than both unmodified and no-BAFF control T cells, while no-BAFF control T cell cytotoxicity was not significantly different from that of unmodified T cells (Fig. 1d). This confirms that our BAFF-CAR construct is functional and that the presence of BAFF at the N-terminus is necessary for cytotoxic function.

**Characterization of BAFF-CAR gene integration into T cells using a non-viral platform**. After characterizing the functionality of the CAR-T cells, we wanted to assess the location of transposon integration as well as the presence of any clonal outgrowth of *TcB* mediated transposon integration events. Historical data of viral integration[32,33], as well as 2 other transposase systems[34], were compared to transposon integration using *TcB* (Fig. 2a). After analysis of all integrants, the results indicated that *TcB* inserted the transposon into transcript-coding DNA regions less frequently than lentivirus (Fig. 2a and Supplementary Table 1). When assessing the distance between integration events and transcriptional start sites, transposon-based methods of integration, including *TcB*, integrated further away from transcriptional start sites when compared to lentivirus (Fig. 2b). The cell products generated in this study were compared to available lentivirus data and similarly did not exhibit significant clonal outgrowth (Table 1).

As vector copy number (VCN) is an important consideration for any future regulatory filing, we evaluated transposon copy number via droplet digital PCR (ddPCR). Seven independent

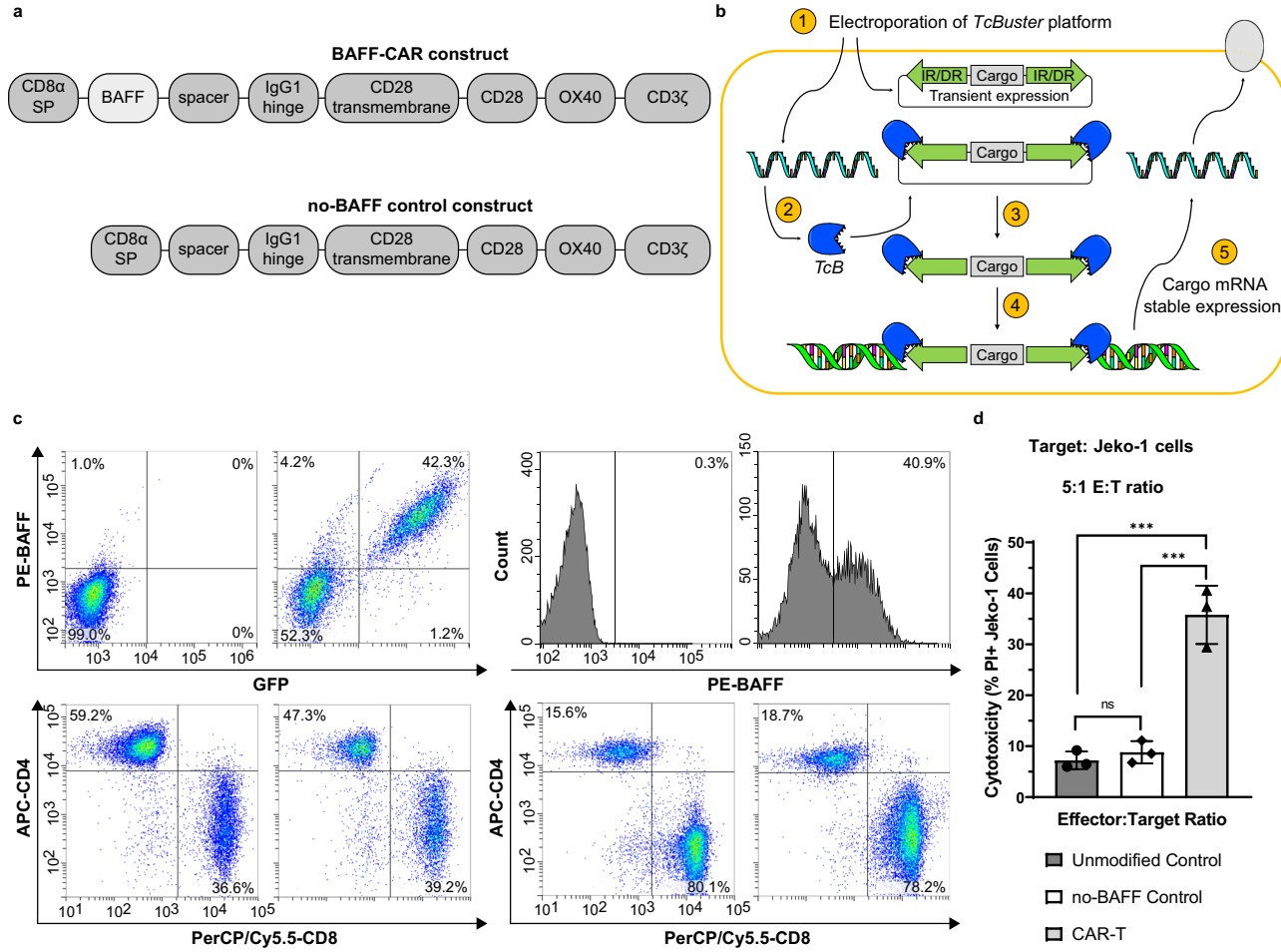

**Fig. 1 CAR-T cells were produced using the BAFF-CAR construct via different expression methodologies. a** Schematic of the BAFF-CAR construct, which consists of extracellular BAFF ligand, short spacer, hinge from human IgG1, CD28 transmembrane and signaling domains, OX40, and CD3ζ. A construct lacking extracellular BAFF (no-BAFF) was used as a negative control. **b** Schematic of the non-viral *TcBuster (TcB)* transposon system, which enables stable expression of a CAR protein. 1. *TcB* transposase mRNA and transposon plasmid are introduced into the cell. 2. Protein from *TcB* mRNA is produced. 3. Transposase cuts the cargo from the transposon plasmid. 4. *TcB* transposase pastes the transposon cargo into the genomic DNA. 5. Cargo mRNA is stably expressed from genomic DNA. **c** The BAFF-CAR construct was expressed in primary human T cells via either lentiviral transduction or the *TcB* transposase system. 5 days after lentiviral transduction, efficiency of transduction was measured based on % GFP expression and correlated exogenous surface expression of BAFF. Efficiency of *TcB* transposase-mediated expression was measured based solely on % exogenous BAFF expression. CD4+ and CD8+ T cell percentages were also measured. The experiment was repeated with four different T cell donors. **d** CAR-T cells, no-BAFF control T cells, or unmodified T cells were co-cultured with fluorescently-labeled Jeko-1 MCL cells at 5:1 effector:target (E:T) ratio for 16 h, followed by flow cytometry. Cytotoxicity was measured via propidium iodide (PI) staining and gating on labeled target cells. ns not significant, ***$P < 0.001$. $P = 0.0002$ for Unmodified Control vs CAR-T, $P = 0.0003$ for no-BAFF Control vs CAR-T. Graphs display mean ± SD, $n = 3$ biologically independent co-cultures, one-way ANOVA with Tukey's multiple comparisons test. The experiment was repeated with two different T cell donors. Source data are provided as a Source Data file.

T cell donors were engineered to express the BAFF CAR with *TcB*. Genomic DNA was harvested and samples were analyzed. No donor achieved a copy number of >10 copies per CAR+ cell (Table 2).

**Characterization of BAFF CAR-T cells**. We next characterized the functionality and specificity of BAFF CAR-T cells. Since soluble BAFF is normally present in human circulation and may be elevated in patients with B cell cancers[24,35], we examined whether it would affect BAFF CAR-T cytotoxicity by competing for receptor binding. Furthermore, soluble forms of all three BAFF receptors have been identified in the blood of healthy adults[36,37]. These levels can change in correlation with autoimmune diseases and cancer, such as chronic lymphocytic leukemia (CLL)[36]. Of the three BAFF receptors, levels of sBAFF-R are highest in healthy adults, though BAFF-R has not been shown

to function as a decoy receptor, in part because it is not believed to be cleaved from the membrane unless it is ligand-bound[36,38]. Given the presence of soluble BAFF ligand and receptors in the blood, and given the potential for sTACI and sBCMA to act as decoy receptors, we performed a spike-in cell killing assay to examine whether these soluble recombinant proteins would inhibit BAFF CAR-T-mediated killing. Physiologically relevant concentrations of each soluble protein were selected, as indicated in the caption of Fig. 3a[36,37]. Results show no diminution of BAFF CAR T cell-mediated cytotoxicity in the presence of any of these soluble BAFF or BAFF receptor family proteins, at any time point or effector:target ratio examined, relative to donor-matched control T cells (Fig. 3a and Supplementary Fig. 2a). We further tested the cytotoxicity of BAFF CAR-T cells in the presence of recombinant BAFF at both pathophysiological (5 ng/mL) and non-physiological, very high (100 ng/mL) concentrations, and we

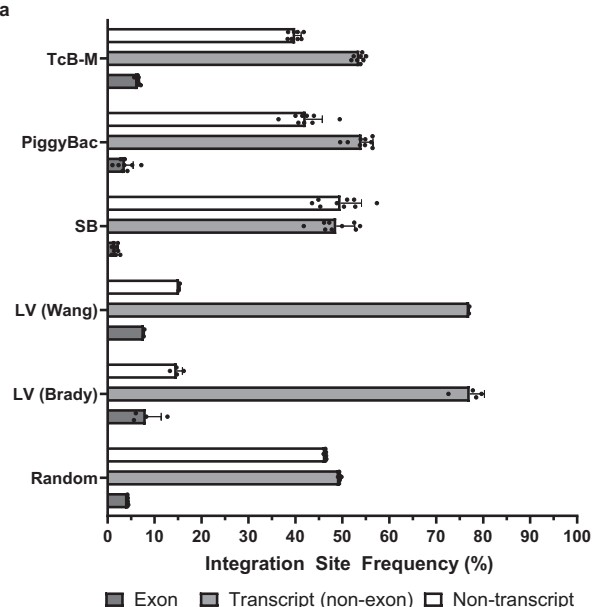

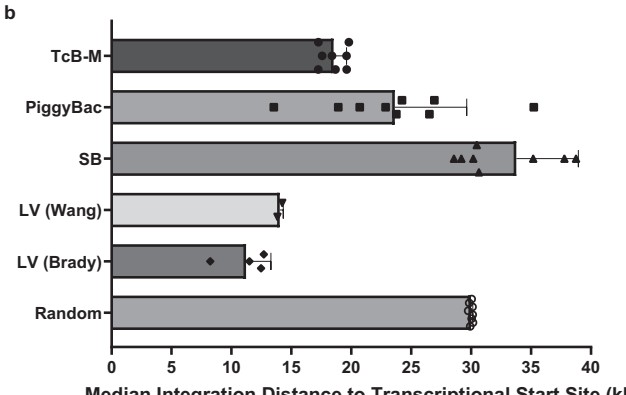

**Table 1 Clonal outgrowth frequency of T cell clones.**

| Gene delivery method | Frequency of top clone | Cumulative frequency of top 10 clones |
|---|---|---|
| *TcB-M* | 0.05% ± 0.02% | 0.41% ± 0.17% |
| *PiggyBac* | 2.52% ± 1.74% | 9.12% ± 2.83% |
| *Sleeping Beauty* | 2.58% ± 1.37% | 13.80% ± 7.76% |
| Lentivirus (Wang) | 0.50% ± 0.25% | 0.96% ± 0.34% |
| Lentivirus (Brady) | 2.17% ± 0.56% | 12.96% ± 1.98% |

Summary table of the frequency of the top 10 T cell clones and the respective percentages of those clones with the T cell population, following different gene delivery conditions. *TcB* transposon system was tested and compared with lentiviral transduction datasets[32,33], as well as *PiggyBac* and *Sleeping Beauty* transposon systems[34]. Table displays mean ± SD, biologically independent samples. $n = 8$ for TcB-M, $n = 9$ for SB, $n = 9$ for PiggyBac, $n = 2$ for LV (Wang), $n = 4$ for LV (Brady). Source data are provided as a Source Data file. Sequencing data used to perform integration site analysis are available online (Accession ID: PRJNA779430).

**Table 2 BAFF-CAR integration copy number in T cells.**

| Donor | Copies/Cell | CAR+% | Copies/CAR+ |
|---|---|---|---|
| D1 CAR | 2.70 | 59.7% | 4.53 |
| D1 Tn Only | −0.80 | 0.0% | 0.00 |
| D10 CAR | 1.11 | 38.5% | 2.88 |
| D10 Tn Only | 0.06 | 0.0% | 0.00 |
| D11 CAR | 3.72 | 58.4% | 6.36 |
| D11 Tn Only | 1.83 | 0.0% | 0.00 |
| D13 CAR | 3.77 | 65.2% | 5.78 |
| D13 Tn Only | −0.85 | 0.0% | 0.00 |
| D15 CAR | 4.19 | 69.1% | 6.06 |
| D15 Tn Only | −0.19 | 0.0% | 0.00 |
| D17 CAR | 3.95 | 59.0% | 6.70 |
| D17 Tn Only | 0.25 | 0.0% | 0.00 |
| D18 CAR | 7.10 | 89.0% | 7.98 |
| D18 Tn Only | 0.44 | 0.0% | 0.00 |

T cells from different donors were transfected with the *TcB* transposon to measure the copy number of the integrated BAFF-CAR coding sequence. The number of copies per T cell and the percentage of CAR+ T cells was used to calculate the number of copies per CAR-T cell. Source data are provided as a Source Data file.

**Fig. 2 Integration site analysis reveals reduced transcript integration with *TcB*. a** Analysis of the frequency of integration events in exon-coding transcript sites, non-exon-coding transcript sites, and non-transcript sites. Samples were collected from primary human T cells transfected with *TcBuster* (*TcB*) or historical data sets generated by Wang et al., Brady et al., and Gogol-Döring et al.[32–34]. Random in silico control sets were also generated and analyzed. SB *Sleeping Beauty*, LV lentiviral dataset. Graph displays mean ± SD, biologically independent samples. $n = 8$ for TcB-M, $n = 9$ for SB, $n = 9$ for PiggyBac, $n = 2$ for LV (Wang), $n = 4$ for LV (Brady), $n = 8$ for Random. **b** Median distance measured between transposon integration sites and the transcriptional start site of the nearest gene is presented for transposon and lentiviral integration. SB *Sleeping Beauty*, LV lentiviral dataset. Graph displays mean ± SD, biologically independent samples. $n = 8$ for TcB-M, $n = 9$ for SB, $n = 9$ for PiggyBac, $n = 2$ for LV (Wang), $n = 4$ for LV (Brady), $n = 8$ for Random. Source data for all graphs are provided as a Source Data file. Sequencing data used to perform integration site analysis are available online (Accession ID: PRJNA779430).

did not observe any changes in cytotoxicity against Jeko-1 cells under either co-culture condition (Supplementary Fig. 2b).

Along with testing the functionality of the BAFF-CAR construct, we also wanted to characterize its specificity. To do this, we used HEK293T cells, which normally do not express any of the BAFF receptors, and engineered them to exogenously express either of the three BAFF receptors (BAFF-R, TACI, or BCMA), verified via flow cytometry (Supplementary Fig. 2c). BAFF CAR-T or unmodified T cells were co-cultured with these modified or

parental HEK293T cells for 16 h at 5:1 E:T ratio. We observed that BAFF CAR-T cells exhibited significantly increased cytotoxicity against all three BAFF receptor-expressing HEK293T cell lines but not against the parental cells, compared to unmodified T cells (Fig. 3b).

Finally, we tested whether BAFF CAR-T cells display any adverse toxicity towards different normal cell types from a number of organ systems. BAFF CAR-T cells or unmodified control T cells were co-cultured with different isolated primary human cell types, including airway epithelial cells, aortic smooth muscle cells, cardiac myocytes, central and peripheral nervous system neurons, hepatocytes, renal epithelial cells, ovarian surface epithelial cells, Sertoli cells, and placental trophoblasts. Adverse toxicity was benchmarked against cytotoxicity against Jeko-1 cells, which was repeated for each type of primary cells. We did not observe any remarkable adverse toxicity against these cell types when compared to unmodified T cells (Fig. 3c), further highlighting the clinical potential of BAFF CAR-T cells.

**In vitro cytotoxicity of BAFF CAR-T cells against multiple B cell cancers**. Next, we comprehensively tested the in vitro cytotoxicity of the BAFF CAR-T cells against several different cell lines that represent different B cell malignancies: Jeko-1 (MCL), rs4;11 (ALL), RPMI-8226 (MM), U266 (MM), and MM.1s (MM) cell lines. These cell lines express one or more of the three BAFF

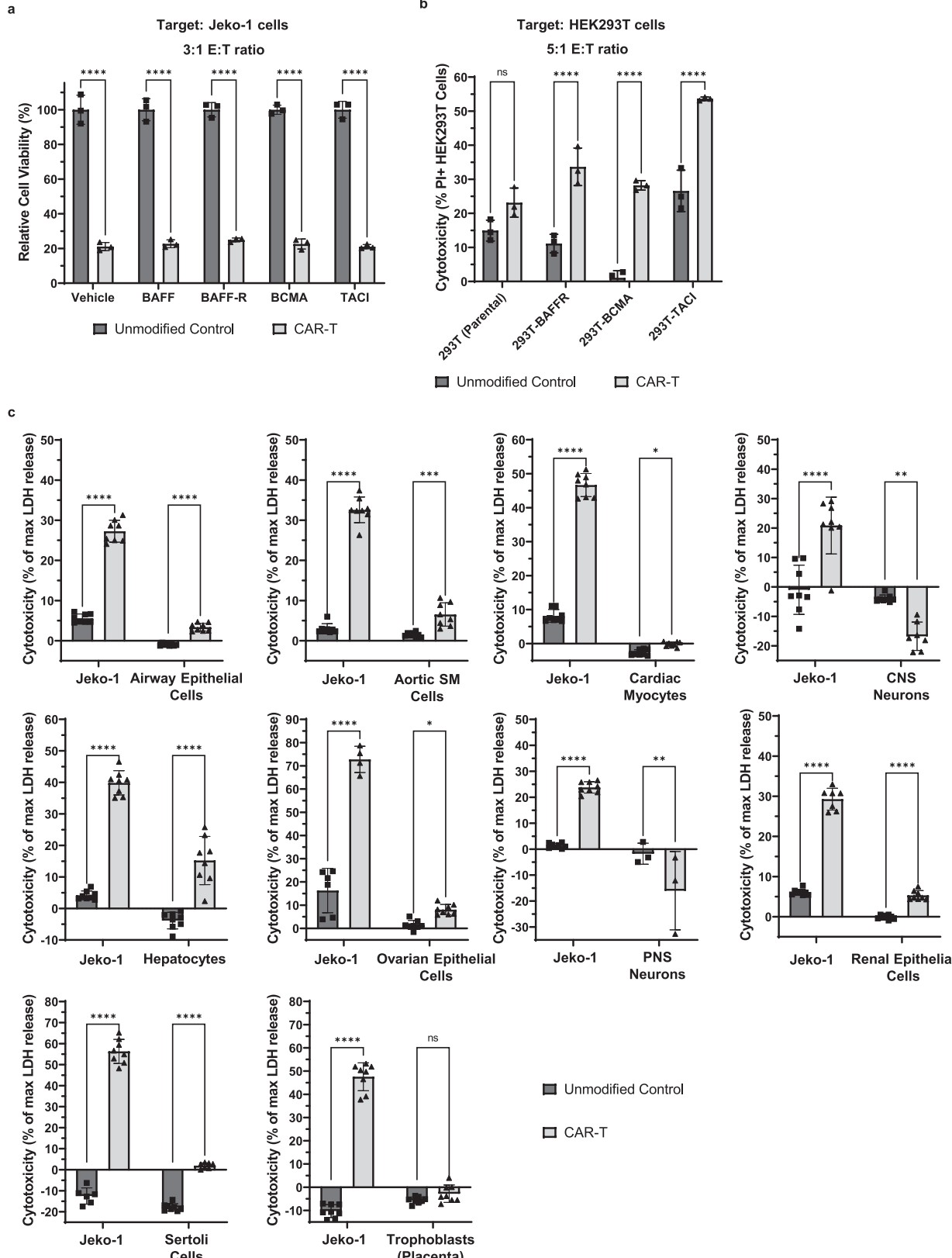

receptors, as evident from flow cytometry analysis (Fig. 4a). Jeko-1 cells are known to express high levels of BAFF-R and low levels of BCMA, whereas the opposite is true for the MM cell lines (Fig. 4a). Both the MCL and MM cell lines express TACI. The rs4;11 cells express relatively high BAFF-R, a moderate amount of TACI, and virtually no BCMA. The MM cells stained negative for CD19 and thus serve as a CD19-negative cancer model for BAFF CAR-T testing.

To measure cytotoxicity, BAFF CAR-T cells were co-cultured with fluorescently-labeled cancer cells at different E:T ratios for either 16 h (Jeko-1, rs4;11, and MM.1s) or 40 h (RPMI-8226 and U266), followed by PI staining to measure cell death via flow

**Fig. 3 BAFF CAR-T cells were characterized and experimentally validated for function and specificity. a** CAR-T cells or unmodified T cells were co-cultured with luciferase-expressing Jeko-1 cells at 3:1 E:T ratio for 24 h with soluble recombinant BAFF/BAFF receptors (sBAFF = 1 ng/mL; sBAFF-R = 500 ng/mL; sBCMA = 2.5 ng/mL; TACI = 20 pg/mL). Vehicle served as a negative control. Luminescence was measured using a plate reader. Viability was calculated using target-only control, then normalized relative to average luminescence from target cells co-cultured with unmodified T cells. ****$P < 0.0001$. Mean ± SD, $n = 3$ biologically independent co-cultures, two-way ANOVA with Šídák's multiple comparison test. The experiment was repeated with three different T cell donors. **b** CAR-T cells or unmodified T cells were co-cultured with HEK293T cells expressing BAFF-R, BCMA, or TACI at 5:1 E:T ratio for 16 h. Cytotoxicity was measured via PI staining and gating on labeled target cells. ns not significant, ****$P < 0.0001$. Mean ± SD, $n = 3$ biologically independent co-cultures, two-way ANOVA with Šídák's multiple comparisons test. The experiment was repeated with two different T cell donors. **c** BAFF CAR-T cells or unmodified T cells were co-cultured with different types of primary human cells at 1:1 E:T ratio for 24 h to evaluate the potential for adverse toxicity. Lactate dehydrogenase (LDH) release upon target cell lysis was measured, then compared with LDH release in lysis controls (0.8% Triton X-100) to calculate cytotoxicity. Percent cytotoxicity = [Experimental LDH release – Effector (spontaneous LDH release) – Target (spontaneous LDH release)]/[Target (maximum LDH release) – Target (spontaneous LDH release)] × 100. CNS central nervous system, PNS peripheral nervous system. ns not significant, *$P < 0.05$, **$P < 0.01$, ***$P < 0.001$, ****$P < 0.0001$. The following p-values correspond to Unmodified Control vs CAR-T: $P < 0.0001$ for airway epithelial cells, $P = 0.0003$ for aortic smooth muscle cells, $P = 0.0377$ for cardiac myocytes, $P = 0.0027$ for CNS neurons, $P < 0.0001$ for hepatocytes, $P = 0.0401$ for ovarian surface epithelial cells, $P = 0.0094$ for PNS neurons, $P < 0.0001$ for renal epithelial cells, $P < 0.0001$ for Sertoli cells, $P =$ not significant for trophoblasts. Mean ± SD, $n = 8$ biologically independent co-cultures, two-way ANOVA with Šídák's multiple comparisons test. The experiment was repeated with two different T cell donors. Source data for all graphs are provided as a Source Data file.

cytometry. BAFF CAR-T cells exhibited significantly enhanced cytotoxicity against all of these B cell malignancies compared to unmodified T cells (Fig. 4b). This was true for all tested cell lines and at all tested E:T ratios. At a 5:1 E:T ratio, Jeko-1 cells (69%) and rs4;11 cells (57%) displayed significant cell death mediated by BAFF CAR-T cells. At a 3:1 E:T ratio, BAFF CAR-T cells displayed the greatest increase in cytotoxicity against MM.1s cells (63%). Similar enhanced cytotoxicity was observed at a 1:5 E:T ratio against RPMI-8226 cells (57%) and U266 cells (37%) (Fig. 4b). At these same ratios, unmodified T cells displayed much lower cytotoxicity against target cell lines. Thus, we have shown that BAFF CAR-T cells produced using non-viral genetic engineering are cytotoxic against multiple B cell cancers.

To confirm the involvement of BAFF/BAFF receptor binding in mediating cytotoxicity and also to mimic the antigen escape scenario in patients, we generated Jeko-1 cells in which either BAFF-R, TACI, or both BAFF-R and TACI receptor expression were knocked out using CRISPR/Cas9. Next, we measured the cytotoxicity of BAFF CAR-T cells against these cells. Jeko-1 cells express high levels of BAFF-R and moderate levels of TACI, although they express virtually no BCMA on the surface (Fig. 4a). Thus, by knocking out both BAFF-R and TACI, we generated Jeko-1 cells with no detectable BAFF receptor expression, serving as a tool to confirm the specificity of the BAFF-CAR construct. Successful knockout cells were verified and separated from untransfected cells via fluorescence-activated cell sorting (FACS) (Supplementary Fig. 3a). As expected, BAFF CAR-T cells displayed significantly reduced cytotoxicity towards Jeko-1 cells with dual BAFF-R/TACI receptor knockout, compared to parental Jeko-1 cells (Fig. 4c). Interestingly, BAFF CAR-T cells were able to kill TACI or BAFF-R single knockout cells effectively, showing that either TACI or BAFF-R expression is enough for BAFF-CAR-mediated killing. Dual BAFF-R/TACI receptor knockout caused a sharp decrease in cytotoxicity compared to TACI knockout alone. This shows that the targeting of both receptors contributes significantly to BAFF-CAR functionality, but Jeko-1 cell expression of TACI alone does not impact BAFF CAR-T activity. Furthermore, loss of either BAFF-R or TACI alone still leaves BAFF CAR-T cells with significant cytotoxic capability.

Similarly, we used CRISPR/Cas9 to knock out either BCMA, TACI, or both BCMA and TACI in RPMI-8226 cells since they express high levels of both BCMA and TACI but no detectable levels BAFF-R. Knockout cells were verified and separated from untransfected cells via FACS (Supplementary Fig. 3b). BAFF CAR-T cells exhibited significantly decreased cytotoxicity against

dual BCMA/TACI knockout cells (Fig. 4d). Single TACI knockout RPMI-8226 cells also showed some inhibition in cytotoxicity, while BCMA knockout cells behaved similar to parental cells. Thus, we have shown that we are able to successfully produce BAFF CAR-T cells that are both functional and specific against target cells that express any of the three BAFF receptors, even when expression of one of the natural receptors is abrogated experimentally.

**BAFF CAR-T cell activation after co-culture with cancer cells.** We next characterized T cell activation by staining for CD69, a T cell activation marker that is expressed on the surface following antigen engagement. BAFF CAR-T cells displayed much higher levels of CD69 expression than unmodified cells when co-cultured with all target cell lines (Fig. 5a). This can be compared to CD69 expression at baseline, without the presence of target cells, in which a much smaller proportion of BAFF CAR-T cells express surface CD69 (~16%), albeit still somewhat higher than with unmodified T cells (~4%) (Supplementary Fig. 4a). We also stained for CD107a, a degranulation marker expressed on the surface of cytotoxic immune cells during the release of lytic enzymes such as perforin and granzyme B. BAFF CAR-T cells exhibited significantly higher levels of CD107a, whereas unmodified cells displayed negligible levels of CD107a on their surface (Fig. 5b). Without the presence of target cells, neither BAFF CAR-T cells nor unmodified T cells exhibit appreciable surface expression of CD107a (Supplementary Fig. 4b).

Next, we used a multiplex, bead-based flow cytometry assay to measure the release of various pro-inflammatory cytokines and lytic enzymes from T cells after co-culture with target cells. Against all tested cancer cell lines, the BAFF CAR-T cells released significantly higher levels of the pro-inflammatory cytokines TNF-α and IFN-γ, lytic enzymes granzymes A and B and perforin, soluble Fas ligand (sFasL), and granulysin (Fig. 5c). In order to measure the baseline release of these cytokines and lytic enzymes, we co-cultured BAFF CAR-T cells and unmodified control T cells with and without Jeko-1 cells. The multiplex assay revealed that BAFF CAR-T cells do not express significantly different levels of TNF-α, IFN-γ, sFasL, or granzyme A compared to control T cells at baseline, but when co-cultured with Jeko-1 cells, the release of these cytokines increased dramatically for BAFF CAR-T cells (Supplementary Fig. 4c). To measure IL-2 secretion from these T cells, we co-cultured BAFF CAR-T cells and unmodified control T cells with Jeko-1 cells in IL-2-free media, and we observed that BAFF CAR-T cells secrete

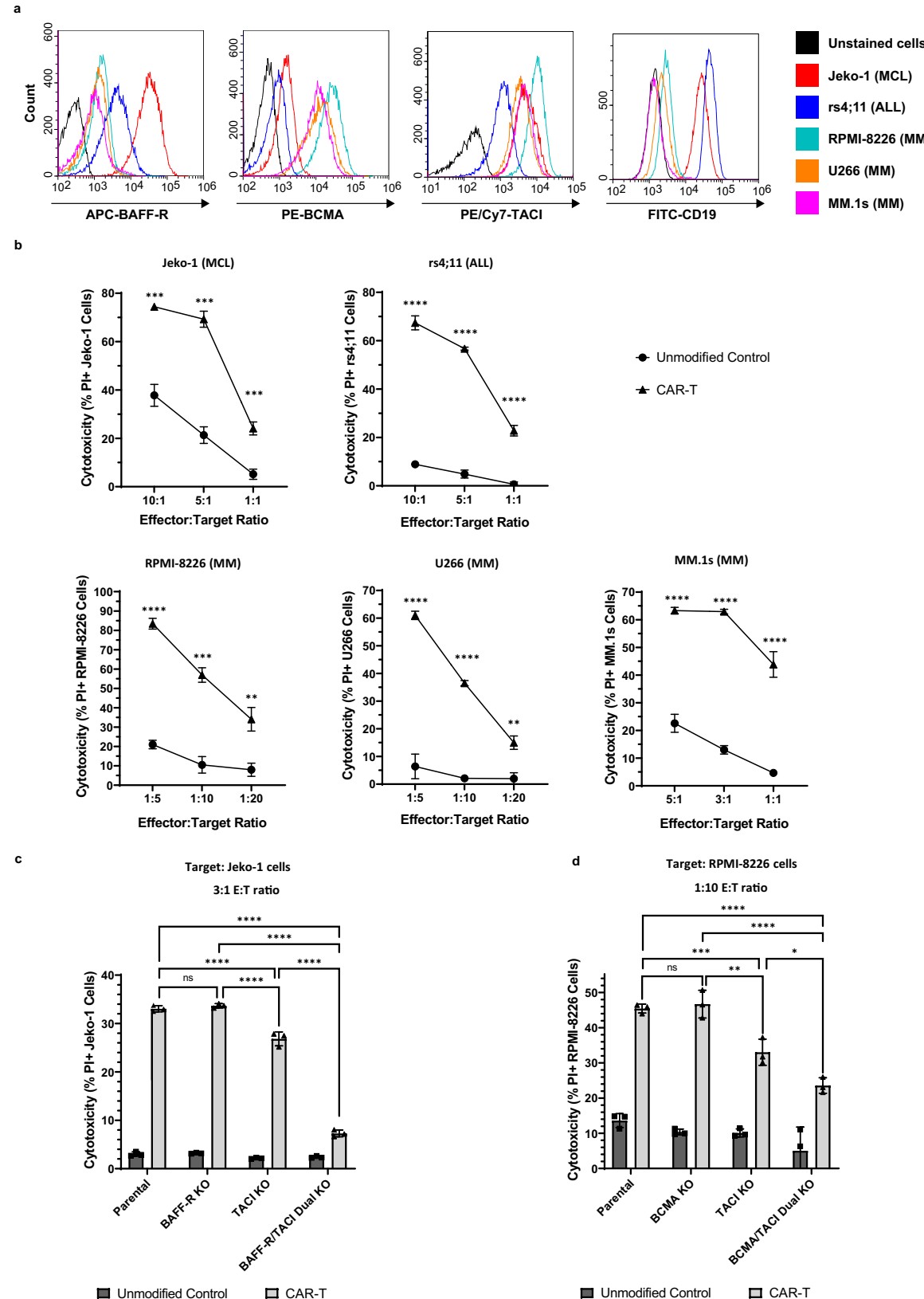

significantly higher levels of IL-2 compared to control (Supplementary Fig. 4d).

**In vivo cytotoxicity of BAFF CAR-T cells against MCL liquid cancer.** We analyzed the effect of BAFF CAR-T against a liquid MCL xenograft model by using Jeko-1 cells that were modified to express the firefly luciferase gene (Jeko-1-luc). Male or female NSG mice were injected intravenously (i.v.) with Jeko-1-luc cells (1.5e6) and on Day 9 post-tumor inoculation, mice were injected i.v. with BAFF CAR-T (10e6) or unmodified Control-T cells (10e6) (Supplementary Fig. 5a). Bioluminescent imaging was conducted weekly until Day 36 (Fig. 6a and Supplementary

**Fig. 4 BAFF CAR-T cells display significant in vitro cytotoxicity towards MCL, ALL, and MM cells. a** Cell lines of different B cell malignancies were stained for BAFF-R, BCMA, TACI, and CD19 expression. Red = Jeko-1 (MCL); blue = rs4;11 (ALL); cyan = RPMI-8226 (MM); orange = U266 (MM); purple = MM.1s (MM); black = unstained cells. **b** CAR-T cells or unmodified T cells were co-cultured with fluorescently-labeled cancer cells at indicated E:T ratios for 16 h (Jeko-1, rs4;11, MM.1s) or 40 h (RPMI-8226, U266). Cytotoxicity was measured via PI staining and gating on labeled target cells. Circle = Unmodified Control, triangle = CAR-T. **$P < 0.01$, ***$P < 0.001$, ****$P < 0.0001$. Jeko-1: $P = 3.34e\text{-}4$ (10:1), $P = 1.92e\text{-}4$ (5:1), $P = 6.72e\text{-}4$ (1:1). rs4;11: $P = 9e\text{-}6$ (10:1), $P = 3e\text{-}6$ (5:1), $P = 8.8e\text{-}5$ (1:1). RPMI-8226: $P = 2.1e\text{-}5$ (1:5), $P = 2.85e\text{-}4$ (1:10), $P = 2.919e\text{-}3$ (1:20). U266: $P = 7.2e\text{-}5$ (1:5), $P = 3e\text{-}6$ (1:10), $P = 2.179e\text{-}3$ (1:20). MM.1s: $P = 6.6e\text{-}5$ (5:1), $P = 3e\text{-}6$ (3:1), $P = 1.29e\text{-}4$ (1:1). Mean ± SD, $n = 3$ biologically independent co-cultures, multiple unpaired two-tailed $t$-tests with Holm-Šídák correction for multiple comparisons. **c** CRISPR was used to knock out expression of either BAFF-R, TACI, or both BAFF-R and TACI in Jeko-1 cells. Fluorescently-labeled cancer cells were co-cultured with BAFF CAR-T cells or unmodified T cells 16 h at 3:1 E:T ratio. Parental Jeko-1 cells served as negative control. Cytotoxicity was measured as above. ns not significant, ****$P < 0.0001$. Mean ± SD, $n = 3$ biologically independent co-cultures, two-way ANOVA with Tukey's multiple comparisons test. **d** CRISPR was used to knock out expression of either BCMA, TACI, or both BCMA and TACI in RPMI-8226 cells. Fluorescently-labeled cancer cells were co-cultured with BAFF CAR-T cells or unmodified T cells 36 h at 1:10 E:T ratio. Parental RPMI-8226 cells served as negative control. Cytotoxicity was measured as above. ns not significant, *$P < 0.05$, **$P < 0.01$, ***$P < 0.001$, ****$P < 0.0001$. $P = 0.0015$ (Parental vs TACI KO), $P < 0.0001$ (Parental vs BCMA/TACI Dual KO), $P = 0.0006$ (BCMA KO vs TACI KO), $P < 0.0001$ (BCMA KO vs BCMA/TACI Dual KO), $P = 0.0137$ (TACI KO vs BCMA/TACI Dual KO). Mean ± SD, $n = 3$ biologically independent co-cultures, two-way ANOVA with Tukey's multiple comparisons test. Source data for all graphs are provided as a Source Data file. All cytotoxicity experiments were repeated with two different T cell donors.

Fig. 5b). Mice treated with Control-T cells were either dead or sick due to extensive tumor burden by Day 36, while BAFF CAR-T-treated mice were all alive and did not show any disease symptoms (Fig. 6a). Bioluminescence signal intensity for BAFF CAR-T-treated mice remained significantly lower compared to control mice over the course of the experiment (Supplementary Fig. 5b). We observed these mice until Day 60 and a survival curve was plotted. BAFF CAR-T-treated mice displayed significantly prolonged survival in contrast to the Control-T-treated mice (Fig. 6b). We wanted to gauge T cell persistence in these mice. Therefore, we collected peripheral blood from several mice in each treatment group on Days 23, 37, 51, and 58 post-innoculation to quantify the number of human CD3+/CD45+ T cells in circulation via flow cytometry (Fig. 6c). After gating on the T cells, we also measured the percentage of cytotoxic CD8+ T cells (Fig. 6d).

There were 2 mice in the CAR-T treatment groups that relapsed, and we wanted to determine whether their relapsed disease was due to antigen loss of any of the BAFF receptors from the surface of the Jeko-1 cells. We processed spleen and bone marrow samples of these mice at the time of euthanasia. Using flow cytometry and gating on human CD19, we did not observe any significant change in the receptor expression profile of Jeko-1 cells from the spleen of 2 relapsed CAR-T-treated mice, as compared to Jeko-1 cells from Control-T-treated mice (Fig. 6e). In addition to assessing the possibility of antigen loss, at the experimental endpoint, spleen and bone marrow samples were collected from BAFF CAR-T-treated mice to compare the percentage of Jeko-1 cells and human T cells in these compartments with previously collected samples from Control-T-treated mice. Representative samples from BAFF CAR-T-treated mice show virtually undetectable Jeko-1 cells in both spleen and bone marrow, as determined by staining for human CD19, whereas the cell population of the spleen and bone marrow of Control-T-treated mice are dominated by Jeko-1 cells (Supplementary Fig. 6). Analogously, BAFF CAR-T-treated mice show high percentages of T cells in the spleen and some bone marrow samples, while Control-T-treated mice tended to have low or undetectable amounts of T cells.

We conducted another experiment using the same i.v. Jeko-1 xenograft model to determine whether a lower dose of CAR-T cells would exhibit similar efficacy. Male or female NSG mice were injected i.v. with Jeko-1-luc cells (1.5e6) and after 1 week, mice were injected i.v. with BAFF CAR-T cells (3e6) or Control-T cells (3e6) or PBS (Supplementary Fig. 7a). Bioluminescent imaging was conducted weekly until Day 70. The PBS and

Control-T treatment group demonstrated aggressive progression of the disease; female mice were killed by Day 28 and most male mice by Day 35 (Supplementary Fig. 7b). However, mice that received the BAFF CAR-T cells showed significantly reduced tumor burden up to Day 70. BAFF CAR-T-treated mice displayed significantly prolonged survival in contrast to the PBS- and Control-T-treated mice (Supplementary Fig. 7c). As expected, the BAFF CAR-T-treated mice had a lower average spleen weight compared to mice treated with PBS or Control-T cells (Supplementary Fig. 7d). Furthermore, BAFF CAR-T-treated mice maintained a relatively steady and healthy weight (Supplementary Fig. 7e), and they did not display any obvious signs of distress or morbidity. We also included a group of mice without tumor inoculation and only injected with BAFF CAR-T cells in order to assess BAFF CAR-T related toxicity. NSG mice were injected i.v. with BAFF CAR-T cells (3e6), and during the 70-day period, they did not exhibit any signs of morbidity from the BAFF CAR-T cells alone, nor did they experience any concerning changes in body weight (Supplementary Fig. 7f). After 70 days, we assessed serum markers for liver toxicity and also collected organs including kidney, liver, lung, and spleen for histological analysis. We measured common liver injury markers alanine aminotransferase (ALT) and aspartate aminotransferase (AST) activity in serum from healthy NSG mice and BAFF CAR-T cell injected mice. We observed no significant differences in ALT or AST activity between the two cohorts (Supplementary Fig. 7g). There were no significant lesions, tissue damage, or notable changes in tissue architecture in BAFF CAR-T injected mice (Supplementary Fig. 8).

**In vivo cytotoxicity of BAFF CAR-T cells against MCL solid, MM liquid, and ALL liquid cancers.** We next wanted to assess the in vivo efficacy of BAFF CAR-T cells using B cell cancer xenograft models. We used a solid MCL model where Jeko-1 cells were injected subcutaneously (s.c.) into immunodeficient NSG mice. After visible tumor development, mice were treated with a single intratumoral injection of either BAFF CAR-T cells, unmodified Control-T cells, or PBS alone (Supplementary Fig. 9a). We observed that while tumor volume continued to increase in mice treated with PBS or Control-T cells, tumor volume decreased rapidly after treatment with BAFF CAR-T cells (Fig. 7a). Mice treated with BAFF CAR-T cells also experienced significantly prolonged survival compared to controls (Fig. 7b).

Next, we tested BAFF CAR-T efficacy using a bioluminescent MM.1s myeloma cell line (MM.1s-luc) to generate an intravenous

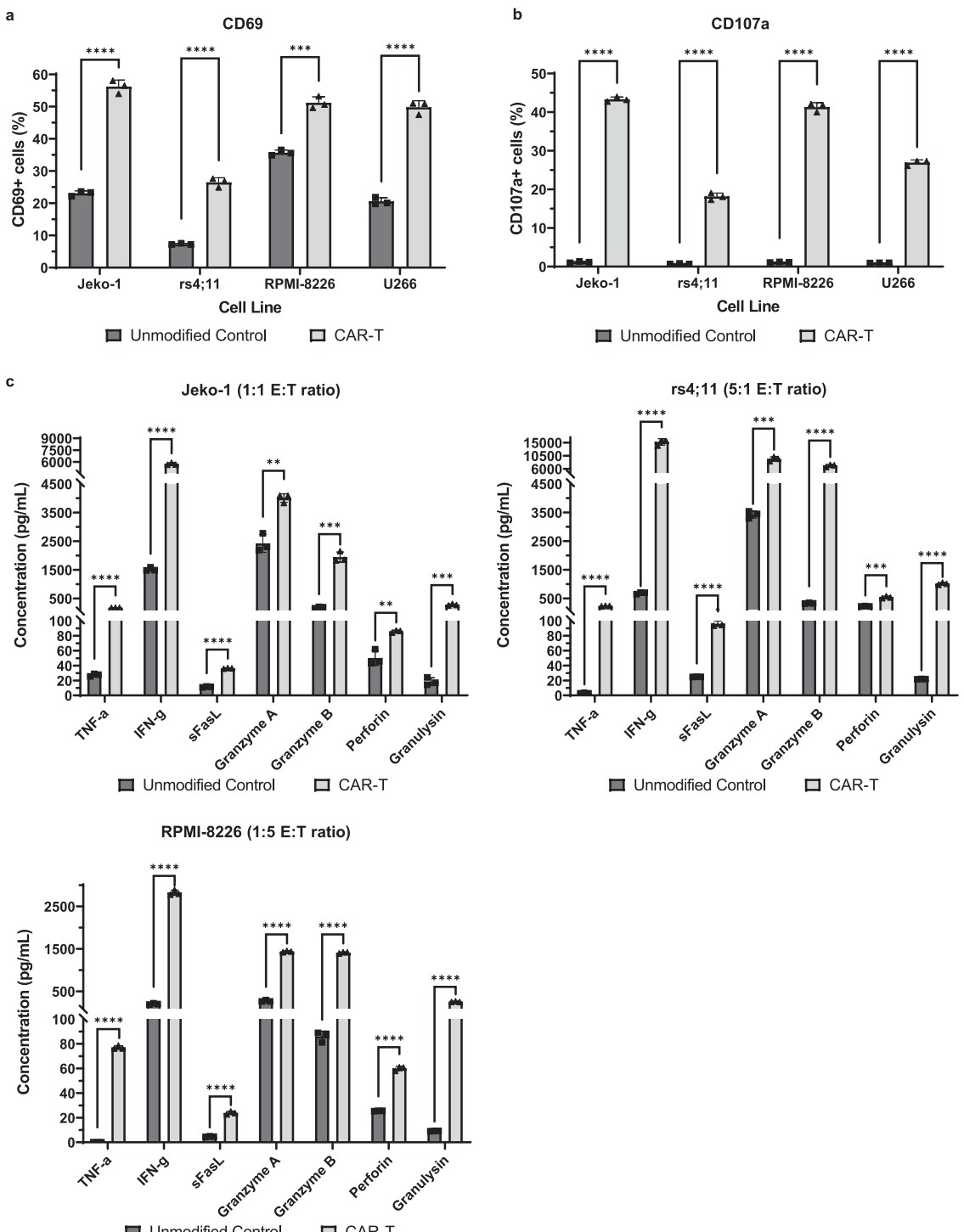

MM xenograft model. MM.1s-luc cells (5e6) were injected i.v. into NSG mice via tail vein, followed by treatment with CAR-T cells (2e6) or Control-T cells (2e6) 8 days and 16 days later (Fig. 7c and Supplementary Fig. 9b). Imaging was performed every 3–5 days until Day 29, when tumor burden was clearly visible in the majority of Control-T-treated mice (Fig. 7c). Bioluminescence intensity was quantified and plotted, showing a clear and significant inhibition of tumor burden in BAFF CAR-T-treated mice compared to Control-T-treated mice (Fig. 7d). BAFF CAR-T-treated mice experienced much slower tumor progression than Control-T-treated mice.

We next tested the BAFF CAR-T cells against two different ALL xenograft models. In one model, rs4;11 cells were injected i.v. into NSG mice, followed by infusion of CAR-T cells or PBS in control mice 6 days later (Supplementary Fig. 9c). Mice were killed 45 days post-tumor inoculation and CD19+ tumor load in blood, spleen, and bone marrow was quantified using flow cytometry. Mice treated with BAFF CAR-T cells showed significantly lower percentages of CD19+ rs4;11 cells (Fig. 7e) compared to control mice. Additionally, the average weight of harvested spleens from these mice was significantly lower, compared to mice treated with PBS alone (Fig. 7f). In addition, we developed a patient-derived

**Fig. 5 BAFF CAR-T cell activation, degranulation, and cytokine release are stimulated by different B cell cancers. a** T cells were co-cultured with labeled Jeko-1 or rs4;11 cells at 5:1 E:T ratio for 24 h, RPMI-8226 cells at 1:5 E:T ratio for 5 h, or U266 cells at 5:1 E:T ratio for 5 h, then stained for the activation marker CD69. % CD69+ T cells were measured via flow cytometry after gating on live cells and excluding labeled cancer cells; for CAR-T samples, additional gating over GFP+ cells was applied to exclude unmodified cells. ***$P < 0.001$, ****$P < 0.0001$. Mean±SD, $n = 3$ biologically independent co-cultures, unpaired two-tailed t-test for each cell line. Experiment was repeated with 2 different T cell donors. **b** T cells were co-cultured with Jeko-1, rs4;11, RPMI-8226, or U266 cells at 5:1 E:T ratio for 6 h while staining for the degranulation marker CD107a. % CD107a+ T cells were measured via flow cytometry after gating on CD3+ cells; for CAR-T samples, additional gating over GFP+ cells was applied to exclude unmodified cells. ****$P < 0.0001$. Mean ±SD, $n = 3$ biologically independent co-cultures, unpaired two-tailed t-test for each cell line. Experiment was repeated with two different T cell donors. **c** T cell and Jeko-1, rs4;11, or RPMI-8226 co-culture supernatant was collected from cytotoxicity assays to measure T cell release of various pro-inflammatory cytokines and lytic enzymes using a multiplex cytokine release assay. **$P < 0.01$, ***$P < 0.001$, ****$P < 0.0001$. For Jeko-1, Unmodified Control vs CAR-T: TNF-α: $P = 1e-6$; IFN-γ: $P = 7e-6$; sFasL: $P = 3e-6$; Granzyme A: $P = 2.751e-3$; Granzyme B: $P = 2.9e-4$; Perforin: $P = 3.834e-3$; Granulysin: $P = 2.9e-4$. For rs4;11, Unmodified Control vs CAR-T: TNF-α: $P = 9e-6$; IFN-γ: $P = 5.9e-5$; sFasL: $P = 2.2e-5$; Granzyme A: $P = 2.19e-4$; Granzyme B: $P = 2.9e-5$; Perforin: $P = 2.19e-4$; Granulysin: $P = 9e-6$. For RPMI-8226: TNF-α: $P < 1e-6$; IFN-γ: $P < 1e-6$; sFasL: $P = 1.1e-5$; Granzyme A: $P < 1e-6$; Granzyme B: $P < 1e-6$; Perforin: $P = 7e-6$; Granulysin: $P = 7e-6$. Mean±SD, $n = 3$ biologically independent co-culture samples, multiple unpaired two-tailed t-tests with Holm-Šídák correction for multiple comparisons. Experiment was repeated with two different T cell donors. Source data for all graphs are provided as a Source Data file.

xenograft (PDX) model using ALL blasts collected from a drug-resistant ALL patient (Pt2 cells). Pt2 cells were injected i.v. into NSG mice and then treated 6 days later (Supplementary Fig. 9d). Mice were killed 31 days post-tumor inoculation. Similar to the results from the rs4;11 model, mice treated with BAFF CAR-T cells had almost negligible counts of CD19 + Pt2 cells remaining compared to mice treated with PBS (Fig. 7g). CAR-T-treated mice also had significantly lower average spleen weight compared to mice treated with PBS alone (Fig. 7h).

## Discussion

Of the CAR-T cell therapies in clinical development, a third still target CD19, the same target as the currently approved CD19-CAR T therapies[19]. Beyond CD19, 70% of these clinically investigated CAR-T therapies are directed against just 10 antigens, reflecting a significant duplication of targets[19]. We have developed a BAFF ligand-based CAR construct and have successfully delivered it into T cells using a non-viral transposon system. BAFF CAR-T targets BAFF-R, BCMA, and TACI expressed on surface of almost all B cell cancers. CD19 CAR-T cell therapy adverse effects include severe B cell aplasia, due to pan-B cell expression of the CD19 marker. Unlike CD19 CAR-T cells, BAFF CAR-T cells designed to target BAFF receptors may produce less severe B cell aplasia, as the BAFF receptors are not expressed by early-stage B cells. Unaffected early B cells could easily replenish the mature B cell population.

Experiments using HEK293T cells as target cells confirmed that the BAFF CAR-T induced cytotoxicity is indeed mediated by BAFF/BAFF receptors interaction. The parental HEK293T cell line does not express any of the BAFF receptors, and as expected, there was no cytotoxicity by BAFF CAR-T cells. Once we transduced these HEK293T cells to overexpress each of the BAFF receptors, these cells were efficiently killed by BAFF CAR-T cells. These data show that the BAFF-CAR construct specifically targets three BAFF receptors and even expression of one of the receptors by target cells is enough to be killed by BAFF CAR-T. This also supports our hypothesis that treatment failure in patients as a result of antigen escape is expected to be much less likely in response to BAFF CAR-T therapy. Another level of confirmation came from the double knockout of BAFF-R and TACI in Jeko-1 cells. Jeko-1 parental cells express both BAFF-R and TACI, and knocking out both of these receptors using CRISPR significantly reduced BAFF CAR-T cell-mediated cytotoxicity. Similarly, double knockout of BCMA and TACI in RPMI-8226 cells also greatly reduced BAFF CAR-T cell-mediated killing. There is some residual cytotoxicity observed in double knockout RPMI-8226 cells and we attribute this to the

low level of BAFF-R expression seen in these cells. Either one of the three BAFF receptor's expression by cancer cells was sufficient to evoke a cytotoxic response by BAFF CAR-T cells. Some primary cancer cells are reported to express all three BAFF receptors[5,39,40], and in those rare cases, BAFF CAR-T will be able to target all three receptors. Most of the B cell cancers express at least two of the three BAFF receptors[41–44], and in those cases it will be dual-targeting. Even if one receptor is lost, the other one is sufficient to evoke a cytotoxic response by BAFF CAR-T cells.

We also observed various degrees of cytotoxicity at the tested E:T ratios against the different cancer cell lines, which express varying levels of the three BAFF receptors. Understanding the binding affinities of BAFF to each of its receptors can also help in estimating the binding affinity of BAFF-CAR towards these endogenous receptors, though the fusion of the CAR backbone may modify it to a certain extent. BAFF has been reported to bind its receptors with varying affinity, with highest affinity towards BAFF-R, followed by TACI and then BCMA[35]. Therefore, even as our multi-antigen targeting BAFF-CAR-T cells may benefit from the stronger binding interactions of BAFF to BAFF-R, they also take advantage of the slightly lower affinity interactions between BAFF and TACI or BCMA, which may confer enhanced efficacy and persistence against a number of different B cell cancers that simultaneously express two or more of these BAFF receptors[41–44].

We have demonstrated the cytotoxicity of BAFF CAR-T cells against MCL, ALL, and MM cells in vitro and in vivo using xenograft models. Each of these in vivo models was done using BAFF CAR-T cells generated from T cells isolated from different donors, thus validating the efficacy of BAFF CAR-T cells generated from multiple donors. It is highly likely that BAFF CAR-T cytotoxicity is not limited to only these cancer cells and that it will serve as a potential therapy for other B cell cancers as well, including those that are not effective using CD19 CAR-T therapy. MM cells do not express CD19, but they express TACI and BCMA; this is a perfect example to show that BAFF CAR-T cells can display significant cytotoxicity against cancer cells that do not express CD19. Though we noticed some minor increases in LDH release after co-culturing BAFF CAR-T cells with normal cells, there were no signs of toxicity noticed in vivo even after 70 days of injecting BAFF CAR-T cells, as evident from body weight, AST/ALT levels, and histology analysis of different organs.

Biologics targeting BAFF/BAFF-R signaling have also been effective for the treatment of autoimmune diseases[45,46], but are still being investigated in clinical trials for cancer (for example, NCT03400176). To our knowledge, using the BAFF ligand as a ligand-based CAR as we describe here has not been previously reported. Relative to a scFv-based BAFF-Receptor CAR[47], which targets only BAFF-R, our preliminary data supports that our non-

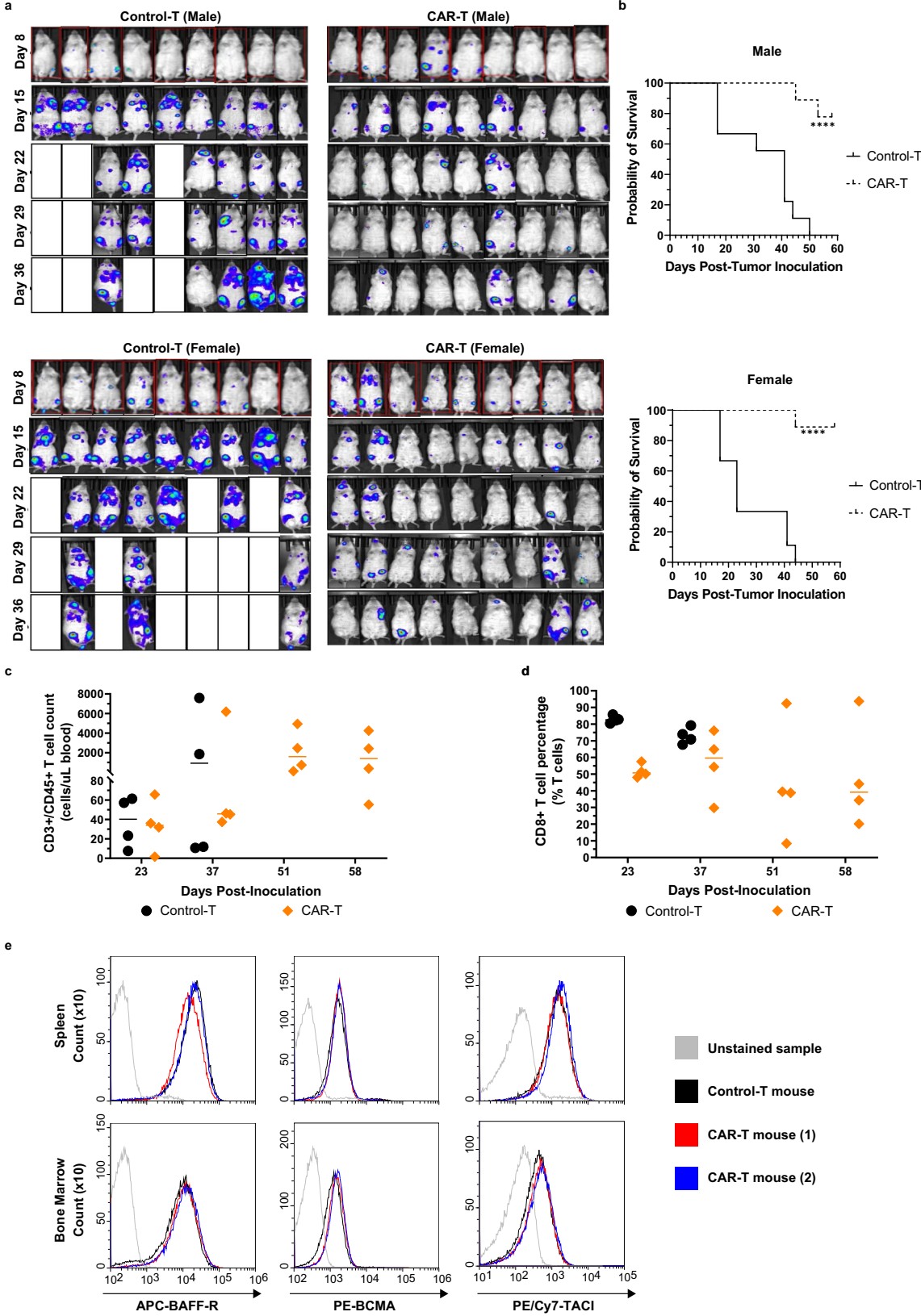

viral engineered BAFF-CAR T cells possess the added benefits of multi-marker targeting and decreased manufacture complexity. Beyond these BAFF-R-CARs, related ligand-based CAR approaches have been reported utilizing a "proliferation-inducing ligand" (APRIL) as an alternative means to BCMA-targeting CAR-T cells for the treatment of MM, which has traditionally

been plagued by antigen escape and low persistence[48]. One of the limitations of our study is that we did not perform a comparison study using our ligand-based BAFF CAR-T v/s existing APRIL CAR-T or scFv CARs such as BCMA CAR-T and BAFF-R CAR-T cells. Hence, we are unable to comment on an efficacy comparison of these CAR-T cells in killing B cell cancers.

**Fig. 6 BAFF CAR-T cells control the progression of cancer cells in an in vivo, intravenous MCL xenograft model. a** NSG mice were injected i.v. via tail vein with 1.5e6 Jeko-1-luc cells on Day 0. On Day 9, 10e6 BAFF CAR-T cells or Control-T cells were injected i.v. Bioluminescence imaging was performed weekly up to Day 36, with the experiment continuing until Day 58. $n = 9$ mice per treatment group. **b** Kaplan–Meier survival curves were generated for male and female mice, with Day 58 post-inoculation serving as the endpoint of the experiment. Solid line = Control-T, dashed line = CAR-T. Statistical analyses of survival between different groups were performed using log-rank (Mantel-Cox) tests. ****$P < 0.0001$ for the pairwise comparison between Control-T and BAFF CAR-T treatment groups, for both male and female mice. **c** Blood was collected from four random mice per treatment group via tail vein on days 23, 37, 51, and 58 post-tumor inoculation, in order to measure T cell, counts using flow cytometry, anti-hCD3, and anti-hCD45 antibodies, and counting beads. Graph displays individual and mean values. Black = Control-T, orange = CAR-T. **d** Human CD8+ T cell percentages over time are graphed as individual and mean values on gated T cells. Black = Control-T, orange = CAR-T. **e** Spleen and bone marrow were collected from relapsed mice at time of euthanasia to assess possibility of antigen loss of BAFF receptors in CAR-T-treated mice. After red blood cell lysis, cells were simultaneously stained for human CD19, BCMA, TACI, and BAFF-R, after which cells were gated on CD19 using flow cytometry. Black = white blood cells from Control-T-treated mice; blue/red = WBCs from two different CAR-T-treated mice with detected Jeko-1 relapse; gray = unstained WBC sample. Source data for all graphs are provided as a Source Data file. The experiment was repeated with two different T cell donors.

In the literature, for some context, there is evidence that the presence of soluble target antigens may reduce CAR efficacy[49–51] and that this may be especially true for sTACI and sBCMA, which can act as decoy receptors for BAFF[52]. Here, our data shows that this is not a concern, as soluble BAFF or BAFF receptors have no effect on BAFF CAR-T cell cytotoxic activity. In our in vivo experiments, MCL cells relapsing after BAFF CAR-T treatment is not due to antigen escape as the MCL cells still express all three BAFF receptors. We hypothesize that it may be due to a decreased population of CAR-T cells in these mice. We did not observe any gender-specific influence on BAFF CAR-T efficacy as both male and female mice treated with BAFF CAR-T showed significantly prolonged survival.

CAR-T cell production using viral transduction is a very common procedure, but there are many disadvantages, including its labor-intensive production and expensive clinical applications. Another serious drawback of using viral vectors is lingering uncertainty over their safety profile. Although viruses have been genetically engineered to reduce their safety risks, there are still concerns of insertional mutagenesis and variable transgene expression. Our analysis of the *TcB* transposon system yielded noteworthy comparisons to both lentiviral vectors and other transposon systems. Overall, *TcB* tended to insert transposons into non-transcript regions much more frequently than lentivirus but somewhat less frequently than the *PiggyBac* (*PB*) or *Sleeping Beauty* (*SB*) transposon systems. *TcB* generally inserted transposons into exonic regions less frequently than lentivirus but somewhat more frequently than *PB* and *SB*. When comparing *TcB*, *PB* and *SB*, *SB* had the highest number of insertions outside of transcribed regions. Finally, when assessing the location of the insertions, it was found that *TcB* inserted DNA farther than transcriptional start sites when compared to lentivirus, but closer to transcriptional start sites when compared to *PB* and *SB*. Additionally, BAFF CAR-T cells created with *TcB* exhibited lower clonal outgrowth than CAR-T cells created using lentiviral, *PB*, or *SB* methods. Thus, *TcB* appears to exhibit an insertion profile that is comparable to existing transposon systems and is potentially advantageous over lentiviral transduction.

Overall, we have demonstrated the effectiveness of a BAFF ligand-based CAR-T cells, generated using non-viral methods, in killing ALL, MCL, and MM cells in vitro and in vivo. We believe these BAFF CAR-T cells have the potential to enter into clinical trials soon for the treatment of multiple B cell cancers.

## Methods

Our research complies with all relevant ethical regulations, as determined by the Case Western Reserve University (CWRU) Institutional Animal Care and Use Committee (IACUC), the CWRU Institutional Biosafety Committee, and the University Hospitals Institutional Review Board (STUDY20191375).

**Cell lines**. All human malignant hematologic cell lines were obtained from the American Type Culture Collection (ATCC), either directly (Jeko-1, CRL-3006;

rs4;11, CRL-1873; MM.1s, CRL-2974) or from colleagues (RPMI-8226, CRM-CCL-155; U266, TIB-196). These cell lines were cultured in complete RPMI-1640 media (10% FBS, 1% penicillin-streptomycin (P/S); Sigma-Aldrich). HEK293T cells were obtained from ATCC (HEK293T/17, CRL-11268) and were cultured in complete DMEM media (10% FBS, 1% P/S; Sigma-Aldrich).

**Flow cytometry**. Flow antibodies (Biolegend) were used at 1/50 dilution to stain cell samples in PBS (Supplementary Table 2). Samples were measured using the Cytoflex flow cytometer (Beckman Coulter). Surface antigen expression (BAFF-R, BCMA, TACI, CD19) was characterized using the Cytoflex flow cytometer (Beckman Coulter) and respective flow antibodies (Biolegend).

**BAFF-CAR construct design**. In order, the BAFF-CAR DNA sequence consists of: CD8α leader sequence; truncated BAFF ligand; short spacer; IgG1 hinge; CD28 transmembrane; CD28; OX40; and CD3ζ domains. The BAFF-CAR DNA sequence was synthesized (Bio Basic Inc.) and subcloned into a modified pLVX lentiviral expression vector (Addgene) using the SpeI and NdeI restriction enzyme sites. The pLVX vector backbone was modified to co-express GFP downstream of the BAFF-CAR construct, separated by a tandem P2A/T2A self-cleaving peptide sequence obtained from the pUltra vector (Addgene).

**T cell isolation and CAR-T production (lentiviral)**. Anonymized healthy donor blood was provided by the CWRU Hematopoietic Biorepository and Cellular Therapy Core under IRB#09-90-195. Peripheral blood mononuclear cells (PBMCs) were isolated from blood via density gradient centrifugation using Ficoll-Paque Plus (Cytiva). T cells were purified from PBMCs using the Mojosort human CD3 T cell isolation kit (Biolegend), which uses an antibody-based negative selection process. Isolated T cells were cryopreserved in 10% DMSO in FBS at 10e6 cells/mL.

To produce lentivirus, 9e6 HEK293T cells were plated on 10-cm plates 24 h prior to transfection. Xtremegene HP Transfection Reagent (Roche) was used to transfect 1.25 µg PMD2.G envelope plasmid, 3.75 µg PSPAX2 packaging plasmid, and 5 µg of construct plasmid per plate. The plate was replaced with fresh media 16 h later, and lentiviral supernatant was collected at 48 h and 72 h post-transfection. Lentivirus was concentrated using Lenti-X concentrator reagent (Takara) and resuspended in complete RPMI-1640 media.

Isolated T cells were thawed and activated with DynaBeads Human T-Activator CD3/CD28 (ThermoFisher) for 48 h in complete RPMI-1640 media supplemented with IL-2 (2 ng/mL), followed by infection with lentivirus using a "spinfection" centrifugation protocol (3480 rpm, 90 min, 25 °C). CAR-T cells were expanded in complete Advanced RPMI-1640 media (10% FBS, 2 mM L-glutamine, 1% penicillin–streptomycin; ThermoFisher), supplemented with IL-2 (2 ng/mL), IL-7 (10 ng/mL), and IL-15 (5 ng/mL). CAR-T cells were expanded until 10 days post-activation and either used for experiments or cryopreserved.

**T cell isolation and CAR-T cell production (transposon)**. Cells from healthy human donors were obtained from Trima Accel leukoreduction system (LRS) chambers. Red blood cells were eliminated by ammonium chloride-based lysis prior to positive immunomagnetic selection for CD4 and CD8 with the EasySep Human CD4 Positive Selection Kit II and the EasySep Human CD8 Positive Selection Kit II (STEMCELL Technologies). Isolated T cells were cryopreserved in CryoStor CS10 (BioLife Solutions) at 20 × 106 cells/mL.

T cells were thawed, then activated with Dynabeads Human T-Activator CD3/CD28 (ThermoFisher) for 36–48 h, after which beads were magnetically separated from T cells. Luminary Therapeutics obtained Tc Buster (TcB) and Hyperactive TcBuster (TcB-M) from Bio-Techne (Minneapolis, MN).

Transfection of *TcB* transposase mRNA and BAFF-CAR transposon plasmid were achieved using electroporation performed with the 4D-nucleofector (Lonza). Electroporation was performed with the 4D-Nucleofector (Lonza, Basel, Switzerland). T cells were centrifuged to remove the culture medium and resuspended in P3 buffer. For stable BAFF-CAR integration, 2.5 µg of BAFF-CAR

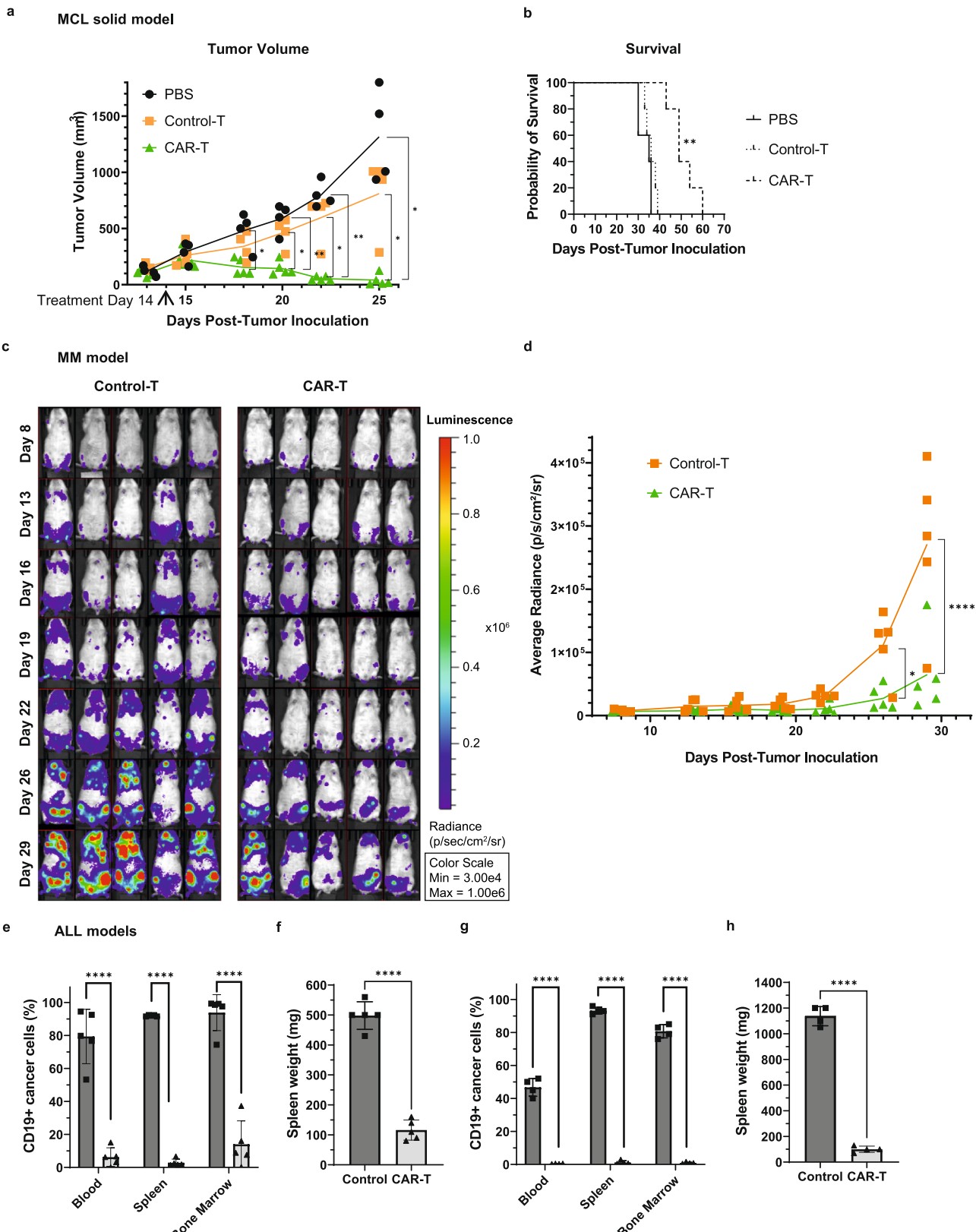

transposon DNA were combined with 5 μg of transposase RNA (BioTechne) per 10e6 cells. T cells were then cultured in CTS OpTmizer T-cell expansion SFM (ThermoFisher) supplemented with 2.5% CTS Immune Cell SR, 2 mM L-glutamine, 10 mM N-acetyl-L-cysteine, 5 ng/mL IL-7, and 5 ng/mL IL-15. Cells were maintained in gas-permeable rapid expansion (G-Rex) culture vessels (Wilson Wolf). All T cells were cultured at 37 °C and 5% $CO_2$.

**Droplet digital PCR for copy number analysis**. Genomic DNA was harvested from expanded T cell samples using the Qiagen QIAamp DNA Blood Mini kit per manufacturer's instructions (Qiagen, Hilden, Germany). Integration PCR was run as a duplexed assay consisting of an internal reference RNAse P primer and probe set (HEX) and an experimental primer and probe set (FAM) designed to target either the promoter of the transgene or the junction between the promoter and the

**Fig. 7 BAFF CAR-T cells display significant in vivo cytotoxicity against MCL, MM and ALL xenograft models. a** 10e6 Jeko-1 cells were injected into NSG mice subcutaneously Day 0 (D0). After palpable tumor formation, 4e6 CAR-T cells, unmodified Control-T cells, or PBS alone were injected intratumorally D14. Tumor volume (mm$^3$) = (length × width$^2$)/2. Black = PBS, orange = Control-T, green = CAR-T. *$P < 0.05$, **$P < 0.01$. D18: $P = 0.0467$ PBS/CAR-T; D20: $P = 0.0068$ PBS/CAR-T, $P = 0.0246$ Control-T/CAR-T; D22: $P = 0.0017$ PBS/CAR-T, $P = 0.0306$ Control-T/CAR-T; D25: $P = 0.0168$ PBS/CAR-T, $P = 0.0421$ Control-T/CAR-T. Mean ± SD, $n = 5$ mice, 2-way ANOVA with Tukey correction for multiple comparisons. **b** Mouse survival post-inoculation. Solid line = PBS, dotted line = Control-T, dashed line = CAR-T. **$P < 0.01$. $P = 0.0054$ Control-T/CAR-T and PBS/CAR-T. Log-rank (Mantel-Cox) tests were applied, followed by Holm-Šídák correction for multiple comparisons. **c** 5e6 MM.1s-luc cells were injected into NSG mice i.v. D0. 2e6 CAR-T or Control-T cells were injected i.v. D8 and D16. Imaging continued until D29 post-inoculation. Luminescence intensity was measured as radiance (photons/s/cm$^2$/sr). **d** Average radiance is plotted for individual mice, with a line connecting means. Orange = Control-T, green = CAR-T. *$P < 0.05$, ****$P < 0.0001$. D26: $P = 0.0128$; D29: $P < 0.0001$. $n = 5$ mice, 2-way ANOVA with Šídák's correction for multiple comparisons. **e** 1e6 rs4;11 cells were injected into NSG mice i.v. D0. 1e6 CAR-T cells or PBS alone (Control) were injected i.v. D6. Mice were killed D45 post-inoculation. Percentage rs4;11 cells remaining were determined using flow cytometry and human anti-CD19 staining. ****$P < 0.0001$. Blood: $P = 1.7e-5$; Spleen: $P < 1e-6$; Bone Marrow: $P = 1.7e-5$. Mean±SD, $n = 5$ mice, multiple unpaired two-tailed $t$-tests with Holm-Šídák correction. **f** Mouse spleen weight from rs4;11 xenograft model after euthanasia. ****$P < 0.0001$. Mean±SD, $n = 5$ mice, unpaired two-tailed $t$-test. **g** 2e6 Patient ALL (Pt2) cells were injected into NSG mice i.v. D0. 2e6 CAR-T cells or PBS alone (Control) were injected i.v. D6. Mice were killed D31 post-inoculation. Percentage Pt2 cells remaining were determined. ****$P < 0.0001$. Blood: $P = 2e-6$; Spleen: $P < 1e-6$; Bone Marrow: $P < 1e-6$. Mean±SD, $n = 4$ mice, multiple unpaired two-tailed $t$-tests with Holm-Šídák correction. **h** Mouse spleen weight from Pt2 xenograft model. ****$P < 0.0001$. Mean±SD, $n = 4$ mice, unpaired two-tailed $t$-test. Source data for all graphs are provided as a Source Data file. Experiments were repeated with two different T cell donors.

transpoton backbone. Reactions were set up using the ddPCR Supermix for Probes (no dUTP) (Bio-Rad, Hercules, CA) with 50 ng of genomic DNA per assay and droplets were generated and analyzed with the QX200 Droplet-digital PCR system (Bio-Rad, Hercules, CA). Frequency was calculated as fractional abundance adjusted for two copies of reference sequence per genome using the QX Manager 1.2 Standard software (Bio-Rad, Hercules, CA).

**Integration site analysis**

*Next-generation sequencing.* Sequencing libraries were prepared from 150 ng genomic DNA quantified by Picogreen (Life Technologies) using the Lotus DNA Library Prep Kit (Integrated DNA Technologies) according to the manufacturer's specifications for libraries undergoing target enrichment. Ligations used vendor-supplied "stubby" adapters, with sample-specific 8-bp unique dual indices (UDIs) added during final library amplification (7 PCR cycles). Hybridization capture was performed per manufacturer's protocol with up to 8 libraries in multiplex (500 ng per library) using xGen universal blocking oligos (IDT) and a custom biotinylated xGen oligo probe pool designed to hybridize to the inserted transposon sequence. Given the small probe panel size, hybridizations and temperature-sensitive washes were performed at 63 °C and the total hybridization time was increased from 4 h to 16 h. Captured libraries were then amplified to ≥2 nM using KAPA HiFi HotStart 2X PCR master mix (11 PCR cycles), quantified by Picogreen, sized on an Agilent TapeStation using the D1000 assay, normalized, and pooled for 150-bp paired-end sequencing on an Illumina NovaSeq* SP flowcell. Sequencing results are available online (Accession ID: PRJNA779430).

*Computational analysis.* We analysed integration site data for *TcBuster* and compared results to published literature data for integration sites of other transposase and viral systems[32–34]. Comparison sequencing datasets were generated by outside sources using different experimental methods. Raw reads from comparison datasets were retrieved (Accession IDs: *Lentivirus*: SAMN11351981, SAMN11351982, SAMN11351983, SAMN11351984, SAMN00188192, SAMN00188193; *Sleeping Beauty*: SAMN02870102; and *PiggyBac*: SAMN02870101) and computationally mapped and analysed in the same manner as the in-house generated data using Python (v3.7.10, run on CentOS 7 Linux) to carry out the multiple steps detailed below.

First, paired reads were merged using BBmerge (v38.89)[53]. Merged reads were trimmed to remove barcodes and primer sequences using Cutadapt (v3.2)[54]. Only sequences containing the expected barcodes, above a minimum length, and mapping to an insert/genome junction were utilized. As a control, to minimize false-mappings, random sequences were generated with lengths and quality scores matched to empirical data. The minimum sequence length after trimming of 25 nucleotides eliminated random sequences mapping to the genome in this size of dataset and was used as the minimum length.

Sequences were then mapped to reference human genome (GRCh38) using Bowtie 2 (v2.4.2)[55]. As a comparison to 'true random' integration, random integration control data were generated in silico by replacing mapped locations from an empirical dataset with randomly generated positions on the same chromosome (random nucleotide value between 1 and the length of the chromosome).

Reads that mapped to the same position (±1 nucleotide to allow for sequencing error) were grouped and tallied as clonal repeats from a single integration event. The tallies of reads mapping to each position were used to estimate the relative abundance of various clones and calculate clonal outgrowth of the top single clone and cumulative of the top ten clones.

Flanking sequences to each mapped read location were retrieved from a local copy of the human genome (GRCh38) and used to view and verify the strongly preferred 'TA' integration site, with the slightly preferred longer site of 'ATnnnCTAGnnnAT', using WebLogo (v0.0.0)[56].

The mapped integration sites were classified based on genomic context using the RefSeq annotations. Annotation bed files were downloaded from the UCSC Table Browser. Mapping was done using the python implementation of BedTools (v0.8.0)[57]. Each mapped read was compared to RefSeq annotations for intron, exon, coding exons, and transcript[58].

**CRISPR knockout of BAFF receptors.** Pooled CRISPR sgRNA (Synthego) targeting the BAFF receptors (BAFF-R, TACI, BCMA) were introduced into luciferase-expressing Jeko-1 cells or RPMI-8226 cells via nucleofection (Lonza Nucleofector 2b). 1ug of each sgRNA was used in tandem with 1.5 μg of Cas9 mRNA (Sigma-Millipore) into 10e6 cells. Percentage knockout efficiency was measured using flow cytometry 5 days following nucleofection. The following sgRNA sequences were used: BAFF-R (TNFRSF13C): CCAGCGCCAGGACCA GUGCC, CAGGGGCAGCGCCGCCUCGC, CGUCCUGGGCGCAGGGCUGC; BCMA (TNFRSF17): GGUGUGACCAAUUCAGUGAA, ACUGAGCUUAAUAA UUUCUU, AAGGACGAGUUUAAAAACAC; TACI (TNFRSF13B): CGCUGUCUCCUGAGCUCUGG, CAGAAGUAUGCACAUUGCUU, GUUC UAUGACCAUCUCCUGA.

**Exogenous BAFF receptor expression in HEK293T cells.** HEK293T cells were transduced with a pLVX lentiviral expression vector coding for either BAFF-R, TACI, or BCMA, as well as a puromycin resistance gene for positive selection. These vectors were created by subcloning the respective coding sequence from cDNA contained in a pCMV3 backbone (Sino Biological). Puromycin selection was performed for 7d (2 μg/mL), and BAFF receptor expression was confirmed using flow cytometry.

**In vitro cytotoxicity (flow-based) assay.** Tumor cells were labeled with eFluor 670 fluorescent dye (Invitrogen) following the manufacturer's protocol, then co-cultured with T cells at different effector:target (E:T) ratios in a 96-well flat-bottom tissue culture plate (Eppendorf). Wells containing tumor cells alone were used as a negative control. Cells were co-cultured in 250 μL complete RPMI-1640 media + IL-2 (2 ng/mL) for 16 or 40 h, depending on the target cell. Soluble recombinant BAFF (5 or 100 ng/mL) was also added in specified experiments. After co-culture, cells were transferred to a 96-well round-bottom plate and centrifuged in order to collect co-culture supernatant for cytokine production analysis. Cells were washed with PBS, then stained with propidium iodide (Cell Signaling) for 20 min at room temperature. Cells were then analyzed using flow cytometry. Tumor cell death was measured by gating on labeled tumor cells in the APC channel and recording the percentage of PI + cells, then subtracting the percentage of PI + cells in the tumor cell-only negative control.

**In vitro cytotoxicity (luciferase-based) assay.** Luciferase-expressing Jeko-1 target cells were co-cultured with T cells in a 96-well plate for 24 h. Soluble recombinant BAFF-R (500 ng/mL), BCMA (2.5 ng/mL), or TACI (20 pg/mL) were added to co-culture media in specified experiments. After co-culture, the remaining luminescence was measured using a plate reader. Wells containing media alone were used to subtract background luminescence. Wells containing target cells only served as a measure of 100% viability. % viability was calculated

by dividing the experimental value over the target-only value, then normalized relative to viability of the negative control specific to each experiment, as indicated in the figure legend.

**In vitro adverse toxicity assay**. Primary human cells were sourced from Lonza (Catalog # R-DRG-505), ScienCell (Catalog # 6110; 6200; 5200; 1520-5; 7310; 4120; 4520), ATCC (Catalog # PCS-301-010), or iXCells Biotechnologies (Catalog # 10HU-214). BAFF-CAR T cells or unmodified donor matched control T cells were incubated in a 96-well plate at a 1:1 effector to target ratio with either the indicated primary human cell type or the Jeko-1 MCL cell line as a positive control. Cell killing was assessed with the CytoTox 96 Non-Radioactive Cytotoxicity Assay kit (Promega), which measures lactate dehydrogenase (LDH) release from cells to determine cytotoxicity. After 24 h, cell killing was assessed per the manufacturer's instructions. Specifically, the following formula was used to calculate cytotoxicity:

$$\text{Percent cytotoxicity} = [\text{Experimental LDH release} - \text{Effector(spontaneous LDH release)} \\ - \text{Target(spontaneous LDH release)}]/[\text{Target(maximum LDH release)} \\ - \text{Target(spontaneous LDH release)}] \times 100$$

Maximum LDH release from Jeko-1 or primary cells was determined by the addition of 0.8% Triton X-100, yielding complete lysis of target cells.

**Multiplex cytokine release assay**. T cell cytokine release was measured using the Legendplex™ multiplex bead-based immunoassay (Biolegend). Co-culture supernatant was obtained from in vitro cytotoxicity assays following centrifugation of cells. The supernatant was further centrifuged at $10,000 \times g$ for 5 min to pellet debris, then transferred to separate tubes and left undiluted or diluted by 1/5 using assay buffer. In all, 25 µL sample was added in triplicate to a V-bottom 96-well plate, followed by addition of 25 µL assay buffer and 25 µL cytokine capture beads. The assay was followed according to manufacturer's protocol.

**CD69 activation assay**. T cells were co-cultured with eFluor 670-labeled tumor cells (5:1 E:T ratio) in complete RPMI-1640 media + IL-2 (2 ng/mL), then labeled with CD69 PC5.5 flow antibody (Biolegend) and analyzed using flow cytometry. To measure the percentage of CD69+ T cells, labeled tumor cells were excluded, and BAFF CAR-T samples were gated on live GFP+ cells, while unmodified T cell samples were gated on live cells (Supplementary Fig. 10).

**CD107a degranulation assay**. T cells were co-cultured with tumor cells (5:1 E:T ratio) in complete RPMI-1640 media + IL-2 (2 ng/mL) containing GolgiStop protein transport inhibitor (BD Biosciences) and CD107a APC flow antibody (Biolegend) for 6 h. Cells were then labeled with CD3 PE antibody (Biolegend) and analyzed using flow cytometry. To measure the percentage of CD107a+ cells, BAFF CAR-T samples were gated on live CD3+ GFP+ cells, while unmodified T cell samples were gated on live CD3+ cells (Supplementary Fig. 10).

**In vivo tumor modeling and CAR-T therapy**

*Animals*. All animal studies were approved by the CWRU IACUC and conducted in accordance with IACUC protocol (2016-0307). NSG mice were obtained either from the CWRU Athymic Animal & Xenograft Core or from The Jackson Laboratory. They were housed in a pathogen-free animal facility maintained by the CWRU Animal Resource Center, either at the Wolstein Research Building or the Small Animal Imaging Facility (SAIF). Animals in each room are observed daily for signs of illness by the animal technician responsible for providing husbandry. Mice were kept in the following environmental conditions: 12-h light/12-h dark cycle, average 20 °C room temperature, 40–60% humidity. As per protocol, for subcutaneous models, tumor volume was continuously measured every 2–3 days until tumor burden reached a maximum of 20 mm in either length or width or until total volume reached 2000 mm³, upon which the animal was killed. For intravenous models, maximal tumor burden was based on the parameters of mouse body condition score, general appearance, and respiration, all of which are given a score based on observation. Animals that reached a tumor burden score of 40 or 20% loss of original body weight were euthanized.

*Subcutaneous models*. Male NSG mice aged 8-12 weeks were challenged with a subcutaneous injection of 10e6 Jeko-1 or RPMI-8226 cells. Once tumor size reached a minimum volume of 70–300 mm³, mice were treated with a single intratumoral injection of PBS, unmodified T cells (Control-T), or CAR-T cells suspended in 200 µL PBS. Tumor volume was continuously measured every 2–3 days until maximal tumor burden was reached, upon which the animal was killed. Tumor volume was measured using the following formula: Volume = (length × width²)/2.

*ALL liquid models*. The patient-derived ALL xenograft model has been previously reported[59], and it was generated as follows. Patient-derived relapsed ALL cells (Pt2) were obtained from a deidentified, discarded blood sample through University Hospitals and the CWRU Hematopoietic Biorepository and Cellular Therapy Core under an IRB exempt approval (STUDY20191375). Pt2 cells were cultured for

1 week on OP-9 stromal cells (ATCC), then injected intravenously into NSG mice for expansion. Mice were killed upon displaying signs of illness ~35–42 days post-transplantation, and a single-cell suspension was obtained from mouse spleen through mechanical disruption and passage through a 70 µm cell strainer. Splenic cells were cultured overnight in MEMα media (Sigma-Aldrich) supplemented with 20% FBS (Sigma-Aldrich), 1% L-glutamine (Gibco), and 1% penicillin/streptomycin (HyClone). Non-adherent cells were confirmed to be >95% human CD19+/CD10+ and were frozen in 95% FBS, 5% DMSO in aliquots for further experimentation.

Male NSG mice aged 8–12 weeks were challenged with an intravenous (i.v.) tail vein injection of rs4;11 (1e6) or Pt2 (2e6) cells. 6 days post-inoculation, mice were injected i.v. with PBS or BAFF CAR-T cells suspended in 200 uL PBS. The health of the mice was monitored per IACUC protocol. 45 days after rs4;11 inoculation or 31 days after Pt2 inoculation, mice were killed and spleen, bone marrow, and blood were harvested for flow cytometry staining for hCD19+ cancer cells and hCD3+ T cells.

*Bioluminescent MM liquid model*. Female NSG mice aged 8–12 weeks were challenged with an i.v. injection of 5e6 MM.1s-luc cells. 2e6 BAFF CAR-T cells or unmodified T cells (Control-T) suspended in 200 µL PBS were injected both 8 days and 16 days post-tumor inoculation. Bioluminescence imaging was performed every 3–5 days, as described below. Health of mice was monitored per IACUC protocol.

*Bioluminescent MCL liquid model*. Stable, luciferase-expressing Jeko-1 cells (Jeko-1-luc) were produced via lentiviral transduction using the pLM expression vector. Male and female NSG mice aged 8–12 weeks were challenged with i.v. injection of 1.5e6 Jeko-1-luc cells via tail vein. Both male and female mice were used for this study, with $n = 5$ mice per treatment group. 7 days post-inoculation, mice were injected with either PBS, unmodified T cells (Control-T), or BAFF CAR-T cells. Bioluminescence imaging was performed weekly, as described below. Estimates of T cell count and the percentage of CD4+ and CD8+ T cells was performed by collecting peripheral blood from the tail vein on Days 23, 37, 51, and 58 post-treatment followed by flow antibody staining and simultaneous red blood cell lysis/sample fixation using Cal-Lyse solution (ThermoFisher). The health of the mice was monitored every 2–3 days and mouse weight was recorded weekly. Animals presenting poor health conditions or significant drop in body weight were euthanized, as per IACUC protocol. Mouse survival was plotted using a Kaplan–Meier curve. Mouse spleen and bone marrow were harvested postmortem, followed by mechanical disruption and passage through a 70 µm cell strainer to obtain a single-cell suspension. These samples were stained for human tumor cells and T cells using flow cytometry. Tumor cell expression of BAFF-R, BCMA, and TACI was determined by gating on human CD19+ cells (Supplementary Fig. 10). Mouse serum was isolated from peripheral blood to measure alanine aminotransferase (ALT) and aspartate aminotransferase (AST) activity using commercial ALT and AST activity assays (Sigma-Aldrich), per manufacturer protocol.

*Bioluminescence imaging*. Mice were anesthetized using isoflurane, followed by an intraperitoneal injection of 150 mg/kg D-luciferin (PerkinElmer). In vivo bioluminescence imaging was performed using an IVIS Spectrum imaging system (PerkinElmer). Luminescence was measured 10 min following D-luciferin injection. Images were analyzed using Living Image software (PerkinElmer).

*Histology*. Mouse kidney, liver, lung, and spleen were harvested and fixed in 4% paraformaldehyde (Biotium) for 6 h at room temperature, then replaced with fresh fixative and stored at 4 °C overnight. After 24 h, fixed tissues were transferred to 70% ethanol and stored at 4 °C. Samples were embedded in paraffin and sectioned, followed by hematoxylin and eosin staining.

**Statistical analysis**. Data were compiled using Microsoft Excel (v2102) and all statistical analyses were conducted using GraphPad Prism v9.2.0 software. Appropriate statistical methods were used to calculate significance, as described in figure legends. Briefly, the following methods were used throughout this study: unpaired two-tailed Student's $t$-test; one-way analysis of variance (ANOVA); two-way ANOVA; and survival analysis using the long-rank Mantel–Cox test. $t$-tests with multiple comparisons were corrected with the Holm-Šídák method, whereas ANOVA analyses with multiple comparisons were corrected using the Tukey, Dunnett, or Šídák tests. The statistical p-value for survival analysis was determined between each pair of treatment groups and corrected for multiple comparisons. In all figures, data are presented as mean ± SD, unless stated otherwise. Experiments were conducted at least twice to ensure reproducibility. For in vivo experiments, the number of mice per treatment group (n) are indicated in figure legends.

## Data availability
The NGS sequencing data generated in this study have been deposited at the Sequence Read Archive (SRA) with BioProject ID PRJNA779430. The remaining data are available within the Article, Supplementary Information or Source Data file. Source data are provided with this paper.

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

## Acknowledgements

This work was supported by St. Baldrick's scholar award (RP), Case Comprehensive Cancer Centre Innovation award (RP), R21 CA246194 (RP), The Case Technology and Validation Program (RP), and the T32 GM007250 training grant (DPW). This research was supported by the Athymic Animal and Hemopoietic Biorepository and Cellular therapy core facilities, Shared Resources of the Case Comprehensive Cancer Center (P30CA043703) for their support. We thank the flow cytometry core and the small animal imaging core facilities at Case Western Reserve University.

## Author contributions

D.P.W. and N.K.R. performed the experiments, analyzed data, and wrote the manuscript, K.Z. performed one in vivo experiment, Anusha Anukanth performed CRISPR experiments, Abhishek Asthana involved in BAFF CAR-T synthesis, N.S. and S.D. performed non-viral gene delivery of CAR constructs and performed some of the in vitro experiments, B.J.J., W.S.L., B.R.W., and B.S.M. performed transposon integration analysis study, P.C. contributed to study design, discussions, helped in troubleshooting and writing the manuscript. R.P. conceived the study, oversaw the study, designed experiments, guided research personnel, performed data analysis, and wrote the manuscript.

## Competing interests

N.S. and S.D. are employees of Luminary Therapeutics. R.P. is a consultant for Luminary therapeutics. B.J.J. is an employee at Bio-Techne. B.R.W. and B.S.M. are consultants for Bio-Techne and are shareholders of Luminary Therapeutics. P.C. is a member of advisory board of ADC therapeutics and have received honoraria from kite pharma. The remaining authors declare no competing interests.
