## [Peer Review File · Nature Communications]

A BAFF ligand-based CAR-T cell targeting three receptors and multiple B cell cancersReviewers' comments:

Reviewer #1 (Remarks to the Author): with expertise in transposon systems

The report by Wong et al. uses the TcBuster transposon system to generate BAFF CAR T cells directed against B cell malignancies. They demonstrate effective cell transfection as well as in vitro/in vivo activity against multiple (solid and liquid) malignancy models. To this reviewer's knowledge, this is the first use of the TcBuster transposon system to generate CAR T cells. The use of a CAR targeting multiple receptors on malignant cells seems like a plus for CAR T directed cancer therapies. The results against multiple B cell malignancies using their CAR T cells seem impressive. Nonetheless, there is important data missing from the manuscript and the methods as described will not permit testing for reproducibility by other scientists.

Major comments:

- 1) The production of the transposon modified CAR T cells is not described. Electroporation and culture conditions are not described at all. This makes the study basically non-reproducible.
- 2) The full sequence of the BAFF CAR transposon construct should be published.
- 3) Which version of the TcBuster transposase was used for these studies?
- 4) What is the phenotype of these CAR T cells? What percent is CD4+, CD8+, etc.?
- 5) They show 40% transfection, but what is the stable integration rate using TcBuster in T cells? If the cells are cultured longer (than 5 days), what percent remain CAR +ive? Without this data, it is impossible to determine the stable integration rate using TcBuster compared to other methods.
- 6) Why is the lentivirus transduction rate of the T cells so low? This is remarkably low compared to what people have reported (and used in patients) for lentivirus and human T cells.
- 7) The number of integrations mediated by TcBuster in T cells has to be reported. This is particularly important given the use of the transposon system and recent reports by Bishop et al., and Micklethwaite et al. in Blood in 2021 of high integrants with transposons used for CAR T cells and resultant cancer development.

Minor comments:

- 1) Dot plots for bar graphs showing individual data points should be included throughout the manuscript and not just select figures. Dot plots should also be included for the tumor growth plots.
- 2) The authors should map genomic TcBuster transposon integration sites in T cells. This is needed for safety assessment compared to other systems.

Reviewer #2 (Remarks to the Author): with expertise in CAR-T

Wong et al reported a valid idea that B cell-activating factor (BAFF) binds to three receptors (BAFF-R, BCMA, and TACI) and the BAFF ligand-based CAR has the potential to minimize the potential antigen escape and reduce the cost and complexity of multiple CAR-T productions in clinical trials.

General Comments:

The study was designed to provide evidence that BAFF CAR can target 3 receptors, BCMA, TACI, and BAFF-R by using HEK293T cells overexpressing the three targets separately. However, data from Fig 2d and Supplemental Figure 1d did not support that the BAFF CAR can efficiently kill TACI and BCMA cells as compared with mock T cells presented in Supplemental Figure 1d at the same E: T ratios. Supplemental Figure 1d should combine with Figure 2d to show the non-specific killing. The only efficient cytotoxicity is against BAFF -R. Although BAFF CAR T cells exhibited killing against different B cell and multiple myeloma cell lines, none of them exclusively expresses only one target. Thus, the data did not support the conclusion that BAFF CAR T cells are capable of targeting three receptors.

Specific Comments:

1. If figure 2b used 5ng/ml BAFF as physiological concentration, it is not clear why 1ng/ml BAFF was used in Figure 2c.
2. Engineering HEK293T cells expressing TACI was not successful based on Supplemental Figure 1c.
3. Figure 4c: has autocrine IL-2 been measured? It would be an important indicator for CAR T cell expansion and persistence.
4. Figure 5. Lack of analysis of BAFFR, TACI, and BCMA expression on the relapsed tumor. It would be nice to present CAR T cell persistence in vivo as well. There are successful MCL and MM models with intravenous injection that can better recapitulate the diseases. Subcutaneous tumor and intratumor CAR treatment are quite artificial and not favorable.
5. Figure 6 a: Tumor burden before treatment on day 7 should be presented as a baseline to ensure that mice have a similar tumor burden to start with. Tumors seem to relapse after transient tumor regression. How to justify the potency of BAFF CAR as compared with existing e.g BAFF-R CAR or BCMA CAR?

6. Figure 6 d: are these CAR gated CD4 or CD8 T cells or total CD4 and CD8 T cells? T cells from the control T group should be added.

Reviewer #3 (Remarks to the Author): with expertise in CAR-T, B cell malignancies

The results of this manuscript are interesting, but there are a number of issues:

Novelty issues:

1. CAR T cell production with non-viral transposon gene delivery method has already been shown for other CARs (SLAMF7 CAR T cells)
2. BAFF CAR T cells and their potential benefit to use in patients escaping CD19 CAR T cell therapy have been earlier published by the same group.
3. The group now claims that these CARs -because they target BAFF-R, BCMA and TACI at once, would diminish the chance of treatment failure as a result of antigen escape also in multiple myeloma. This strong claim is not really shown in this paper. The claim is also curious because these three targets are not expressed at the same time on myeloma cells. Specifically, BAFF-R is strongly downregulated in multiple myeloma cells. Thus it is possible that BAFF-R targeting by BAFF CARs is irrelevant in the multiple myeloma setting.
4. I agree that targeting myeloma cells via both BCMA and TACI could have some benefit, but this is concept is not new, as it has been shown several years ago with the use of APRIL CARs.

Experimental issues :

5. In figure 4, comparison of CART cells with untransduced control is not sufficient. What is the baseline expression of CD69 (Figure 4A) CD107a (figure 4B) and cytokines (figure 4 C,D,E) on unstimulated CART cells?
6. The MM model shown in figure 5c is a very inferior SC model. 11 days of tumor follow up is not good enough to make good conclusions. There are much better xenograft models (NSG mice) in which the tumors can be monitored for longer periods by BLI. In some PDX models the MM tumor can even be grown in a human BM-like niche generated by MSC coated ceramic scaffolds. This latter model could also give the possibility to use patient MM cells derived from BCMA CAR-resistant patients.
7. Although many authors do not pay attention to it, in all in vivo models, intratumoral injection of CART cells poses a serious problem of interpretation. In vivo testing of CART cells should be performed by i.v injections, unless there is a very specific reason not to do it.

8. Finally to understand whether BAFF CARs can target BAFF-R in MM the authors should use BCMA and TACI deficient MM cells.

Safety issues:

9. It may be possible that targeting B cells as well as plasma cells induce deeper immune deficiency in the humoral immune system. Furthermore TACI is also expressed in the myeloid lineage (monocytes, macrophages, dendritic cells). Targeting these cells might have consequences for innate and T cell immune system. These potential safety issues needs discussion, preferentially, after being addressed experimentally.

We thank all three reviewers for their valuable time in reviewing this manuscript and for giving us their constructive suggestions helped to improve the quality of this manuscript. New data has been included as Fig.1c, Fig.2 (a-d), Fig. 3b, Fig.3c, Fig.4c, Fig.4d, Fig.6 (a-e), Fig.7c, Fig.7d, Supplementary figures 1, 2, 3, 4, 5 and 6. We strongly believe that we answered all concerns raised by reviewers and request you to consider this article for publication in Nature Communications.

Please see below point by point answers to reviewers concerns in *bold italics*.

Reviewer #1 (Remarks to the Author): with expertise in transposon systems

The report by Wong et al. uses the TcBuster transposon system to generate BAFF CAR T cells directed against B cell malignancies. They demonstrate effective cell transfection as well as in vitro/in vivo activity against multiple (solid and liquid) malignancy models. To this reviewer's knowledge, this is the first use of the TcBuster transposon system to generate CAR T cells. The use of a CAR targeting multiple receptors on malignant cells seems like a plus for CAR T directed cancer therapies. The results against multiple B cell malignancies using their CAR T cells seem impressive. Nonetheless, there is important data missing from the manuscript and the methods as described will not permit testing for reproducibility by other scientists.

Major comments:

1) The production of the transposon modified CAR T cells is not described. Electroporation and culture conditions are not described at all. This makes the study basically non-reproducible.

All details regarding T cell isolation, electroporation, culture conditions and production of transposon modified CAR-T cells are now included in Materials and Methods of the revised manuscript.

2) The full sequence of the BAFF CAR transposon construct should be published.

Full sequence is publically available as this is protected by a patent and will be also available on request from corresponding author.

3) Which version of the TcBuster transposase was used for these studies?

We used the newly developed hyperactive Tc Buster™ (TcB-M obtained from Bio-Techne). Added this information to Materials and Methods.

4) What is the phenotype of these CAR T cells? What percent is CD4+, CD8+, etc.?

CD4/CD8 percentages are added in Figs. 1c and S1b on this revised manuscript.

5) They show 40% transfection, but what is the stable integration rate using TcBuster in T cells? If the cells are cultured longer (than 5 days), what percent remain CAR +ive?

Without this data, it is impossible to determine the stable integration rate using TcBuster compared to other methods.

Fig. S1c added on this revised manuscript shows stability of BAFF-CAR expression on 5 days and 21 days post-thaw. This data shows the stable integration using TcBuster in T cells.

6) Why is the lentivirus transduction rate of the T cells so low? This is remarkably low compared to what people have reported (and used in patients) for lentivirus and human T cells.

We optimized lentiviral transduction protocol further and achieved a transduction efficiency >40% and is included in Fig. 1c on this revised manuscript. Details of lentiviral transduction of T cells is added in Materials and Methods.

7) The number of integrations mediated by TcBuster in T cells has to be reported. This is particularly important given the use of the transposon system and recent reports by Bishop et al., and Micklethwaite et al. in Blood in 2021 of high integrants with transposons used for CAR T cells and resultant cancer development.

We performed an extensive study to analyze transposon integration sites of TcBuster-M in T cells and compared it to other transposons such as sleeping beauty and piggybac transposons as well as two published lentiviral constructs. Average unique number of insertions is provided in Fig. 2a, Median integration distance to transcription site is shown in Fig. 2b, cumulative frequency of top 10 clones in Fig. 2c, copy numbers per cell in Fig. 2d and number of integrations for individual donors shown in Supplementary Table 1. Insertions in exons, transcript (non-exon), and non-transcript regions are distinguished.

Minor comments:

1) Dot plots for bar graphs showing individual data points should be included throughout the manuscript and not just select figures. Dot plots should also be included for the tumor growth plots.

We plotted dot plots showing individual mouse data points for all in vivo experiments including tumor growth plot. Graphs for in vitro experiments are generated following standard procedures displaying mean +/- Standard Deviation with technical triplicates (n=3) and a 2-way ANOVA with Šídák's multiple comparisons test performed for evaluating statistical significance.

2) The authors should map genomic TcBuster transposon integration sites in T cells. This is needed for safety assessment compared to other systems.

Integration site analysis and copy number analysis were performed and data added as Fig. 2 and Supplementary Table 1. This data includes median distance from transcriptional start sites.

Reviewer #2 (Remarks to the Author): with expertise in CAR-T

Wong et al reported a valid idea that B cell-activating factor (BAFF) binds to three receptors (BAFF-R, BCMA, and TACI) and the BAFF ligand-based CAR has the potential to minimize the potential antigen escape and reduce the cost and complexity of multiple CAR-T productions in clinical trials.

General Comments:

The study was designed to provide evidence that BAFF CAR can target 3 receptors, BCMA, TACI, and BAFF-R by using HEK293T cells overexpressing the three targets separately. However, data from Fig 2d and Supplemental Figure 1d did not support that the BAFF CAR can efficiently kill TACI and BCMA cells as compared with mock T cells presented in Supplemental Figure 1d at the same E: T ratios. Supplemental Figure 1d should combine with Figure 2d to show the non-specific killing. The only efficient cytotoxicity is against BAFF -R.

We generated HEK293T overexpressing BAFF-R, BCMA and TACI receptors and performed cytotoxicity experiments co-culturing with BAFF CAR-T and unmodified T cells, comparing with parental HEK293T cells. Data is added as Figure 3b, including both CAR-T and unmodified T cells in the same figure. Data shows significant cytotoxicity against all HEK293T cells overexpressing BAFF-R, BCMA and TACI.

Although BAFF CAR T cells exhibited killing against different B cell and multiple myeloma cell lines, none of them exclusively expresses only one target. Thus, the data did not support the conclusion that BAFF CAR T cells are capable of targeting three receptors.

Jeko-1 cells expressed predominantly BAFF-R and TACI and showed very low BCMA expression. Hence we generated single knockout of BAFF-R and TACI as well as double knockout of BAFF-R and TACI in Jeko-1 cells using CRISPR technology. We performed cytotoxicity experiments using these cells and added data as Fig.4c. BAFF CAR-T cells were able to kill single knockout Jeko-1 cells very effectively, while double knockout cells were not killed by BAFF CAR-T. This shows even in the absence of BAFF-R or TACI, BAFF CAR-T still able to kill those cancer cells.

We performed similar CRISPR knockouts using RPMI-8226 cells. These cells express predominantly BCMA and TACI, while very low BAFF-R expression was detected. We generated BCMA and TACI single knockout cells as well as double knockout cells and performed cytotoxic experiments and data added in Figs. 4d. Data shows that single knockout cells are being killed by BAFF CAR-T cells, while cytotoxicity of double knockout cells was significantly reduced.

Specific Comments:

1. If figure 2b used 5ng/ml BAFF as physiological concentration, it is not clear why 1ng/ml BAFF was used in Figure 2c.

This figure is moved as Supplementary Figure 2b. 5 ng/mL was used representing pathophysiological concentration of BAFF as seen in autoimmune or cancer patients. 1 ng/mL BAFF concentration represents physiological BAFF serum levels in normal individuals. This information has been added to the manuscript.

2. Engineering HEK293T cells expressing TACI was not successful based on Supplemental Figure 1c.

We generated these HEK293T cells overexpressing BAFF-R, BCMA and TACI again and as shown in Supplementary Figure 2c, these cells were overexpressing BAFF-R (99.8%), BCMA (98.2%) and TACI (74.4%). Please see the cytotoxicity experiment data using these cells in Fig.3b.

3. Figure 4c: has autocrine IL-2 been measured? It would be an important indicator for CAR T cell expansion and persistence.

Autocrine IL-2 is added as Supplementary Figure 4d, after repeating the experiment in IL-2-free media.

4. Figure 5. Lack of analysis of BAFFR, TACI, and BCMA expression on the relapsed tumor. It would be nice to present CAR T cell persistence in vivo as well. There are successful MCL and MM models with intravenous injection that can better recapitulate the diseases. Subcutaneous tumor and intratumor CAR treatment are quite artificial and not favorable.

We performed in vivo experiment using intravenous injection of luciferase labelled MCL cells into male and female NSG mice and tumor intensity monitored by imaging every week after CAR-T cell injection. Imaging data is shown in Fig. 6a, survival data shown in Fig.6b, CAR-T persistence on days 23, 37, 51 and 58 analyzed using CD3 and CD8 staining as shown in Fig.6c, 6d. Lastly, we also analyzed BAFF-R, BCMA and TACI expression on the spleen and bone marrow cells from 2 relapsed BAFF CAR-T treated mice and compared to cells from control-T treated mice (Fig.6e). This data shows there was no antigen escape in these relapsed mice and they still express those receptors. We also measured CAR-T persistence and tumor cells in spleen and bone marrow of these mice and representative data shown in Supplementary Figure S6.

We also performed Multiple Myeloma in vivo model by intravenous injection of luciferase labelled MM1.S cells and data added as Fig.7c and Fig.7d.

5. Figure 6 a: Tumor burden before treatment on day 7 should be presented as a baseline to ensure that mice have a similar tumor burden to start with. Tumors seem to relapse after transient tumor regression. How to justify the potency of BAFF CAR as compared with existing e.g BAFF-R CAR or BCMA CAR?

Baseline tumor burden on day 8 shown in new MCL experiment in Figure 6. T cells were injected on Day 9.

In MM experiment in Fig.7, baseline image on Day 8 is shown.

BAFF-R CAR targets cancer cells expressing BAFF-R only (not TACI and not BCMA). BCMA CAR targets only BCMA expressing cancer cells (not BAFF-R and not TACI expressing cells). Thus, these CAR's target single receptor, while our ligand based BAFF CAR-T targets three receptors (BAFF-R, TACI and BCMA).

6. Figure 6 d: are these CAR gated CD4 or CD8 T cells or total CD4 and CD8 T cells? T cells from the control T group should be added.

We have repeated the invivo experiment using MCL cells.

Added data as Fig.6.

CD3 positive cell counts/ul is shown in Fig.6c and CD8+ percentages (as a percent of total T cells) is show in in Fig.6d.

Reviewer #3 (Remarks to the Author): with expertise in CAR-T, B cell malignancies

The results of this manuscript are is interesting, but there are a number of issues:
Novelty issues:

1. CAR T cell production with non-viral transposon gene delivery method has already been shown for other CARs (SLAM-F7 CAR T cells)

Yes, we agree that non-viral transposon gene delivery has been tried by other groups for CAR-T generation. Here, in this article, novelty is using TcBuster-M transposon for BAFF CAR-T. We anticipate less cost in future clinical trials using BAFF CAR-T generated using Tc-Buster-M clinical grade transposon. A BAFF ligand based CAR-T is also novel and this is the first report.

2. BAFF CAR T cells and their potential benefit to use in patients escaping CD19 CAR T cell therapy have been earlier published by the same group.

We never published this work earlier and this is the first manuscript reporting our novel ligand based BAFF CAR-T. We presented this work in the American Society of Hematology (ASH) in 2020 conference and the abstracts got published in the journal "Blood".

<https://ashpublications.org/blood/article/136/Supplement%201/31/473893/Ligand-Based-CAR-T-Cell-Targeting-BAFF-Receptors>.

This is our first report on BAFF CAR-T capable of targeting multiple receptors and that is the novelty.

3. The group now claims that these CARs -because they target BAFF-R, BCMA and TACI at once, would diminish the chance of treatment failure as a result of antigen escape also in multiple myeloma. This strong claim is not really shown in this paper. The claim is also curious because these three targets are not expressed at the same time on myeloma cells. Specifically, BAFF-R is strongly downregulated in multiple myeloma cells. Thus it is possible that BAFF-R targeting by BAFF CARs is irrelevant in the multiple myeloma setting.

BAFF CAR-T targets cells expressing BAFF-R or BCMA or TACI or combination of them. It is not that it targets only when all three are expressed together. We prove that in 2 ways. First we over expressed these receptors one by one in HEK293 cells (Parental HEK293 cells do not express these receptors). We generated HEK293T overexpressing BAFF-R or BCMA or TACI receptors and performed cytotoxicity experiments co-culturing with BAFF CAR-T and unmodified T cells, comparing with parental HEK293T cells. Data is added as Figure 3b and it shows significant cytotoxicity against all HEK293T cells overexpressing BAFF-R, BCMA or TACI.

Second way of proving this was by knocking out BAFF receptors from cancer cells. Jeko-1 cells expressed predominantly BAFF-R and TACI and showed very low BCMA expression. Hence we generated single knockout of BAFF-R and TACI as well as double knockout of BAFF-R and TACI in Jeko-1 cells using CRISPR technology. We performed cytotoxicity experiments using these cells and added data as Fig.4c. BAFF CAR-T cells were able to kill single knockout Jeko-1 cells very effectively, while double knockout cells were not killed by BAFF CAR-T. This shows even in the absence of BAFF-R or TACI, BAFF CAR-T still able to kill those cancer cells.

We performed similar CRISPR knockouts using RPMI-8226 cells. These cells express predominantly BCMA and TACI, while very low BAFF-R expression was detected. We generated BCMA and TACI single knockout cells as well as double knockout cells and performed cytotoxic experiments and data added in Figs. 4d. Data shows that single knockout cells are being killed by BAFF CAR-T cells, while cytotoxicity of double knockout cells was significantly reduced.

4. I agree that targeting myeloma cells via both BCMA and TACI could have some benefit, but this is concept is not new, as it has been shown several years ago with the use of APRIL CARs.

APRIL CARs target TACI and BCMA, but not BAFF-R.

BCMA CARs target only BCMA, not BAFF-R and not TACI.

BAFF-R CAR's target only BAFF-R, not BCMA and not TACI.

Here we report ligand based BAFF CAR targeting BAFF-R, BCMA and TACI.

Experimental issues :

5. In figure 4, comparison of CART cells with untransduced control is not sufficient. What is the baseline expression of CD69 (Figure 4A) CD107a (figure 4B) and cytokines (figure 4 C, D, E C on unstimulated CART cells?

Baseline expression data for CD69, CD107 and cytokines are included in Supplementary Figure 4. For comparison, Jeko-1 is used for each experiment and shown in same graphs.

6. The MM model shown in figure 5c is a very inferior SC model. 11 days of tumor follow up is not good enough to make good conclusions. There are much better xenograft models (NSG mice) in which the tumors can be monitored for longer periods by BLI. In some PDX models the MM tumor can even be grown a human BM- like niche generated by msc coated ceramic scaffolds. this latter model could also give the possibility to use patient MM cells derived from BCMA CAR-resistant patients.

We performed Multiple Myeloma in vivo model by intravenous injection of luciferase labelled MM1.S cells and data added as Fig.7c and Fig.7d.

7. Although many authors do not pay attention to it, in all in vivo models, intratumoral injection of CART cells poses a serious problem of interpretation. In vivo testing of CART cells should be performed by i.v injections, unless there is a very specific reason not to do it.

We performed in vivo experiment using intravenous injection of luciferase labelled MCL cells into male and female NSG mice and tumor intensity monitored by imaging every week after CAR-T cell injection. Imaging data is shown in Fig. 6a, survival data shown in Fig.6b, CAR-T persistence on days 23, 37, 51 and 58 analyzed using CD3 and CD8 staining as shown in Fig.6c, 6d. Lastly, we also analyzed BAFF-R, BCMA and TACI expression on the spleen and bone marrow cells from 2 relapsed BAFF CAR-T treated mice and compared to cells from control-T treated mice (Fig.6e). This data shows there was no antigen escape in these relapsed mice and they still express those receptors. We also measured CAR-T persistence and tumor cells in spleen and bone marrow of these mice and representative data shown in Supplementary Figure S6.

We also performed Multiple Myeloma in vivo model by intravenous injection of luciferase labelled MM1.S cells and data added as Fig.7c and Fig.7d.

8. Finally to understand whether BAFF CARs can target BAFF-R in MM the authors should use BCMA and TACI deficient MM cells.

As we mentioned it earlier, BAFF CAR-T can target not only BAFF-R, it targets BCMA and TACI as well and that is the novelty of this CAR-T that it can target three different receptors. We performed CRISPR knockouts in RPMI-8226 cells. These cells express predominantly BCMA and TACI, while very low BAFF-R expression was detected. We generated BCMA and TACI single knockout cells as well as double knockout cells and performed cytotoxic experiments and data

added in Figs. 4d. Data shows that single knockout cells are being killed by BAFF CAR-T cells, while cytotoxicity of double knockout cells was significantly reduced. This shows that even if BCMA is lost due to antigen escape, it will work if TACI is present. Similarly, if TACI is lost from MM cancer cells, BAFF CAR-T will be able to kill them as long as there is BCMA present. This BAFF CAR-T can target cancer cells expressing any of these three receptors.

Safety issues:

9. It may be possible that targeting B cells as well as plasma cells induce deeper immune deficiency in the humoral immune system. Furthermore TACI is also expressed in the myeloid lineage (monocytes, macrophages, dendritic cells). Targeting these cells might have consequences for innate and T cell immune system. These potential safety issues needs discussion, preferentially, after being addressed experimentally.

We performed an extensive in vitro toxicity study using many different cells including epithelial, aortic SM cells, cardiac myocytes, CNS neurons, hepatocytes, ovarian epithelial cells, PNS neurons, renal epithelial cells, sertoli cells and trpophoblasts (placenta) and data shown in Fig. 3c. No significant toxicity was noticed against these cells.

Sincerely
Reshmi Parameswaran
09-30-2021

REVIEWER COMMENTS

Reviewer #1 (Remarks to the Author):

The authors of Wong et al. have submitted a revised manuscript including additional data and figures. Regarding novelty, it is difficult to tell if the TcBuster system that is used is any better than other transposon systems already used in clinical trials or is this just another option. As it was obtained from a company, there is no way for other researchers to try and reproduce the data. The additional data demonstrate killing of cells with each of the 3 antigens targeted. However, additional safety issues are raised.

Comments:

1) The description of the comparison of TcBuster integration sites to other platforms is incomplete. The authors only discuss in comparison to lentivirus but not piggyBac or sleeping beauty. For instance, TcBuster appears to target transcription start sites closer than SB.

2) The discussion of adverse toxicity in other cell types is unbalanced. Several of the cell types showed statistically elevated levels of LDH release (airway, aortic SM, hepatocytes, and ovarian). It is difficult to compare in vitro toxicity to what might be observed in vivo. For instance, did the authors check AST/ALT (liver enzymes) in mice treated with CAR-T cells? Did the authors look at any of these tissues where they observed toxicity in vitro?

3) Although multiple different tumoral models are tested, I agree with reviewers 2 and 3 that intra-tumoral injection of CAR-T cells is a poor model.

4) NGS sequencing data should be deposited in NCBI to be available for others.

5) Overexpression in HEK293 (which greatly overexpress) may not correlate with tumor cell line expression.

Reviewer #2 (Remarks to the Author):

The authors have performed additional experiments and the manuscript has undergone extensive revisions. This manuscript by Wong et al has significantly been improved.

The following minor points should be addressed:

1. Fig. 6D meant to address CAR T cell persistence in vivo. However, CD8+ T cells gated from human T cells (CD3+CD45+ in Fig 6C) could be infused CAR negative T cells unless the input cells were 100% CAR. Based on the CAR T cell characterization shown in Supplemental Fig 1, only 50% of the T cells are CAR+. Therefore, data analysis/presentation based on CAR tag (GFP?) is informative. Please clarify it in the Figure legend.

2. Are the data in Fig. 6 and Fig. 7 from one experiment respectively? Replicates with CAR T cells from different donors are needed for in vivo potency assay.

3. Authors should discuss the limitations in the study for not comparing the potency of the existing scFv CARs such as BCMA CAR and BAFF-R CAR vs. ligand-based CAR for specific disease types.

Reviewer #3 (Remarks to the Author):

I am now satisfied with the additional experimental details provided to my questions. The only remaining issue is the positioning of this CAR in the field.

This CAR can "in theory" target three receptors. That is experimentally shown. No problem there.

But in the clinical setting this CAR will never use the power of being able to target three different receptors because in each disease situation (lymphoma or myeloma) one of the targets will always be absent on tumor cells i.e: in lymphoma the BCMA, and in myeloma BAFF-R will be absent. Thus, in the real life this CAR will target only two antigens: BCMA and TACI when used for myeloma treatment and BAFF-R and TACI when used in lymphoma treatment.

Therefore the general claim that a "ligand based BAFF CAR-T which is capable of binding three different receptors, will minimize the potential for antigen escape" is actually not based on triple targeting but only on dual targeting. Therefore, when it comes to positioning of this CAR in the "battlefield" of other competitive CARs, especially in myeloma setting, this CAR will -expectedly- not to achieve a better effect

than earlier described APRIL CARs, which can also target both BCMA and TACI. I think this issue is not trivial but important and therefore needs to be discussed concisely to enable the reader with a balanced view. Important to provide the reader with the message that this this CAR is novel because it can be used in different B cell malignancies to target two different antigens. Triple targeting will not be possible in any clinical setting.

We are happy to address all these concerns raised by reviewers.

Please see below our responses to reviewer's comments in bold. Details added to manuscript is in italics bold.

REVIEWER COMMENTS

Reviewer #1 (Remarks to the Author):

The authors of Wong et al. have submitted a revised manuscript including additional data and figures. Regarding novelty, it is difficult to tell if the TcBuster system that is used is any better than other transposon systems already used in clinical trials or is this just another option. As it was obtained from a company, there is no way for other researchers to try and reproduce the data. The additional data demonstrate killing of cells with each of the 3 antigens targeted. However, additional safety issues are raised.

We do not claim TcBuster is better than other systems. It is just another option and it is available to all researchers from Bio--Techne through an MTA.

Comments:

1) The description of the comparison of TcBuster integration sites to other platforms is incomplete. The authors only discuss in comparison to lentivirus but not piggyBac or sleeping beauty. For instance, TcBuster appears to target transcription start sites closer than SB. **We added few sentences in the discussion section comparing TcBuster to SB and piggyBac transposons.**

“Our analysis of the TcBuster transposon system yielded noteworthy comparisons to both lentiviral vectors and other transposon systems. Overall, TcBuster tended to insert transposons into non--transcript regions much more frequently than lentivirus but somewhat less frequently than PB or SB. TcBuster generally inserted transposons into exonic regions less frequently than lentivirus but somewhat more frequently than PB and SB. When comparing TcB, PB and SB, SB had the highest number of insertions outside of transcribed regions. Finally, when assessing the location of the insertions, it was found that TcBuster inserted DNA farther than transcriptional start sites when compared to lentivirus, but closer to transcriptional start sites when compared to PB and SB. Additionally, BAFF CAR--T cells created with TcBuster exhibited lower clonal outgrowth than CAR--T cells created using lentiviral, PB, or SB methods. Thus, TcBuster appears to exhibit an insertion profile that is comparable to existing transposon systems and is potentially advantageous over lentiviral transduction”.

2) The discussion of adverse toxicity in other cell types is unbalanced. Several of the cell types showed statistically elevated levels of LDH release (airway, aortic SM, hepatocytes, and ovarian). It is difficult to compare in vitro toxicity to what might be observed in vivo. For instance, did the authors check AST/ALT (liver enzymes) in mice treated with CAR--T cells? Did the authors look at any of these tissues where they observed toxicity in vitro?

We agree with the reviewer that the in vitro toxicity assay is just an artificial system and cannot predict what happens in vivo. We added in vivo mouse body weight changes upto 70 days after BAFF CAR--T injection (Supp.Fig 7e, f) AST/ALT levels

(Supp.Fig 7g) and histology of Liver, lungs, spleen and kidney (Supp.Fig 8). Please see results section--

“We also included a group of mice without tumor inoculation and only injected with BAFF

CAR--T cells in order to assess BAFF CAR--T related toxicity. NSG mice were injected i.v. with

BAFF CAR--T cells (3e6), and during the 70--day period, they did not exhibit any signs of morbidity from the BAFF CAR--T cells alone, nor did they experience any concerning changes in body weight (Fig. S7f). After 70 days, we assessed serum markers for liver toxicity and also collected organs including kidney, liver, lung and spleen for histological analysis. We measured common liver injury markers Alanine aminotransferase (ALT) and aspartate aminotransferase (AST) activity in serum from healthy NSG mice and BAFF CAR--T cell injected mice. We observed no significant differences in ALT or AST activity between the two cohorts (Fig. S7g). There were no significant lesions, tissue damage, or notable changes in tissue architecture in BAFF CAR--T injected mice (Fig. S8)”.

3)Although multiple different tumoral models are tested, I agree with reviewers 2 and 3 that intra-tumoral injection of CAR--T cells is a poor model.

Out of the 6 in vivo experiments performed, only one is a sub--cutaneous model with intra-tumoral injection of CAR--T cells. All other 5 in vivo experiments are systemic models with intravenous injection of CAR--T cells. Reviewers 2 and 3 raised this concern initially and we already addressed this in the revised manuscript by including additional in vivo experiments where tumor cells and CAR--T cells are injected intravenously. Only Figure 7a,b is a subcutaneous model of Jeko--1 with intra--tumoral injection of CAR--T. We also demonstrated efficacy of BAFF CAR--T against Jeko--1 using an i.v. model of luciferase-expressing Jeko--1 and intravenous injection of BAFF CAR--T cells in Figure 6 and Supplementary Figure 7. Other in vivo experiments using ALL xenograft, ALL PDX MCL xenograft and MM xenograft are i.v. systemic models with intra--venous CAR--T injections.

4) NGS sequencing data should be deposited in NCBI to be available for others.
Yes, we are happy to deposit sequences in NCBI.

[editorial note] Sequencing data are available at: <https://www.ncbi.nlm.nih.gov/bioproject/779430>

5) Overexpression in HEK293 (which greatly overexpress) may not correlate with tumor cell line expression.

Exogenous expression of BAFF receptors in HEK293 cells was only meant to serve as one way of initial validation of BAFF--CAR functionality and specificity against each individual BAFF receptor. We performed many other experiments with tumor cell lines and CRISPR-modified tumor cell lines (Figures 1d, 3a, 4, 5) to truly demonstrate the functionality and specificity of the BAFF--CAR construct against actual B cell malignancies.

Reviewer #2 (Remarks to the Author):

The authors have performed additional experiments and the manuscript has undergone extensive revisions. This manuscript by Wong et al has significantly been improved.

The following minor points should be addressed:

1. Fig. 6D meant to address CAR T cell persistence in vivo. However, CD8+ T cells gated from human T cells (CD3+CD45+ in Fig 6C) could be infused CAR negative T cells unless the input cells were 100% CAR. Based on the CAR T cell characterization shown in Supplemental Fig 1, only 50% of the T cells are CAR+. Therefore, data analysis/presentation based on CAR tag (GFP?) is informative. Please clarify it in the Figure legend.

We totally agree with this point and in the figure 6d legend it is mentioned as “T cells” based on anti-human CD3 and CD8 staining. As the injected cells contain a mixture of CAR⁺ and CAR⁻ cells, we can only say “T cells” detected instead of CAR--T cells.

2. Are the data in Fig. 6 and Fig. 7 from one experiment respectively? Replicates with CAR T cells from different donors are needed for in vivo potency assay.

CAR--T cells generated using T cells isolated from 6 different donors were used for in vivo experiments. Each in vivo experiment is performed with a different donor. Figures 6 and 7 are also performed with different donors. We used T cells from 6 different donors for performing 6 in vivo experiments including MCL i.v. model (fig.6), MCL solid model (fig.7a, b), Multiple Myeloma model (fig. 7c, d), ALL xenograft (Fig7e, f) model, ALL PDX model (Fig7g, h) and MCL i.v model (Supp.Fig 7). Thus we validated the in vivo efficacy of BAFF CAR--T cells using T cells generated from 6 different donors. We added a sentence in the discussion to make this point clear.

“Each of these in vivo models was done using BAFF CAR--T cells generated from T cells isolated from different donors, thus validating the efficacy of BAFF CAR--T cells generated from multiple donors”.

3. Authors should discuss the limitations in the study for not comparing the potency of the existing scFv

CARs such as BCMA CAR and BAFF--R CAR vs. ligand--based CAR for specific disease types.

We agree with this point made by the reviewer and added this sentence to the discussion. “One of the limitations of our study is that we did not perform a comparison study using our ligand based BAFF CAR--T v/s existing APRIL CAR--T or scFv CARs such as BCMA CAR--T and BAFF--R CAR--T cells. Hence, we are unable to comment on an efficacy comparison of these CAR--T cells in killing B cell cancers”.

Reviewer #3 (Remarks to the Author):

I am now satisfied with the additional experimental details provided to my questions. The only remaining issue is the positioning of this CAR in the field.

This CAR can "in theory" target three receptors. That is experimentally shown. No problem there. But in the clinical setting this CAR will never use the power of being able to target three different receptors because in each disease situation (lymphoma or myeloma) one of the targets will always be absent on tumor cells i.e: in lymphoma the BCMA, and in myeloma BAFF-R will be absent. Thus, in the real life this CAR will target only two antigens: BCMA and TACI when used for myeloma treatment and BAFF-R and TACI when used in lymphoma treatment.

Therefore the general claim that a "ligand based BAFF CAR-T which is capable of binding three different receptors, will minimize the potential for antigen escape" is actually not based on triple targeting but only on dual targeting. Therefore, when it comes to positioning of this CAR in the "battlefield" of other competitive CARs, especially in myeloma setting, this CAR will --expectedly-- not to achieve a better effect than earlier described APRIL CARs, which can also target both BCMA and TACI. I think this issue is not trivial but important and therefore needs to be discussed concisely to enable the reader with a balanced view.

Important to provide the reader with the message that this CAR is novel because it can be used in different B cell malignancies to target two different antigens. Triple targeting will not be possible in any clinical setting.

We acknowledge the reviewer's point that, for the majority of myeloma cases, BCMA and TACI are primarily expressed. In these cases, the BAFF-CAR, like the APRIL-CAR, would be "dual targeting" as there are only two receptors on the surface to target. However, although it may be true that the majority of myelomas are BAFF-R-negative, it is not accurate to say that BAFF-R is absent in 100% of myeloma cases. Novak *et al.* looked at primary CD138+ myeloma cells from 3 different patients and observed varying degrees of BAFF-R expression, in addition to BCMA and TACI. Therefore, we believe it is still legitimate to frame our BAFF-CAR in terms of triple targeting, not just dual targeting. Meanwhile, a significant percentage of primary B cell ALL expresses all three BAFF receptors (Maia *et al.*). We understand that the reviewer is pointing out lymphoma and myeloma specifically, but we should again clarify that our BAFF-CAR construct is meant to target B cell malignancies broadly, not just lymphoma or myeloma alone. Therefore, cases of ALL or other leukemias in which all three BAFF receptors are expressed would fall under the classification of "triple targeting" by our construct.

We added a sentence in the discussion by adding new references

"Some cancer cells are reported to express all three BAFF receptors and in those rare cases, BAFF CAR-T will be able to target all three receptors, while most of the B cell cancers express at least two of the three BAFF receptors³⁹⁻⁴² and in those cases it will be a dual targeting. Even if one receptor is lost, the other one is sufficient to evoke a cytotoxic response by BAFF CAR-T cells".

Reshmi Parameswaran

REVIEWERS' COMMENTS

Reviewer #1 (Remarks to the Author):

The manuscript is acceptable for publication.